# Cretaceous Oceanic Anoxic Events prolonged by phosphorus cycle feedbacks

Sebastian Beil[1], Wolfgang Kuhnt[1], Ann Holbourn[1], Florian Scholz[2], Julian Oxmann[2], Klaus Wallmann[2], Janne Lorenzen[1], Mohamed Aquit[3], El Hassane Chellai[4]

[1]Institute of Geosciences, Christian-Albrechts-University, Ludewig-Meyn-Str. 10-14, D-24118 Kiel, Germany
[2]Geomar Helmholtz Centre for Ocean Research Kiel, Wischhofstr. 1-3, D-24148 Kiel, Germany
[3]OCP S.A., Direction de Recherche et Développement, Recherche Géologique, 46300 Youssoufia, Morocco
[4]Department of Geology, Faculty of Sciences Semlalia, Cadi Ayyad University, Marrakech, Morocco

*Correspondence to*: Sebastian Beil ([Sebastian.Beil@ifg.uni-kiel.de](Sebastian.Beil@ifg.uni-kiel.de)) or Wolfgang Kuhnt ([Wolfgang.Kuhnt@ifg.uni-kiel.de](Wolfgang.Kuhnt@ifg.uni-kiel.de))

**Abstract.** Oceanic Anoxic Events (OAEs) document major perturbations of the global carbon cycle with repercussions on the Earth's climate and ocean circulation that are relevant to understand future climate trends. Here, we compare the onset and development of Cretaceous OAE1a and OAE2 in two drill cores with unusually high sedimentation rates from the Vocontian Basin (southern France) and Tarfaya Basin (southern Morocco). OAE1a and OAE2 exhibit remarkable similarities in the evolution of their carbon isotope ($\delta^{13}$C) records with long-lasting negative excursions preceding the onset of the main positive excursions, supporting the view that both OAEs were triggered by massive emissions of volcanic $CO_2$ into the atmosphere. However, there are substantial differences notably in the durations of individual phases within the $\delta^{13}$C positive excursions of both OAEs. Based on analysis of cyclic sediment variations, we estimate the duration of individual phases within OAE1a and OAE2. We identify: (1) a precursor phase (negative excursion) lasting ~430 kyr for OAE1a and ~130 kyr for OAE2, (2) an onset phase of ~390 and ~70 kyr, (3) a peak phase of ~600 and ~90 kyr, (4) a plateau phase of ~1340 and ~200 kyr and (5) a recovery phase of ~380 and ~440 kyr. The total duration of the positive $\delta^{13}$C excursion is estimated at 2700 kyr for OAE1a and 790 kyr for OAE2 and that of the main carbon accumulation phase at 980 and 180 kyr. The long-lasting peak, plateau and recovery phases imply fundamental changes in global nutrient cycles either (1) by submarine basalt-sea water interactions, (2) through excess nutrient inputs to the oceans by increasing continental weathering and river discharge or (3) through nutrient-recycling from the marine sediment reservoir. We investigated the role of phosphorus on the development of carbon accumulation by analysing phosphorus speciation across OAE2 and the mid-Cenomanian Event (MCE) in the Tarfaya Basin. The ratios of organic carbon and total nitrogen to reactive phosphorus ($C_{org}/P_{react}$ and $N_{total}/P_{react}$) prior to OAE2 and the MCE hover close to or below the Redfield ratio characteristic of marine organic matter. Decreases in reactive phosphorus resulting in $C_{org}/P_{react}$ and $N_{total}/P_{react}$ above the Redfield ratio during the later phase of OAE2 and the MCE indicate leakage from the sedimentary column into the water column under the influence of intensified and expanded oxygen minimum zones. These results suggest that a positive feedback loop, rooted in the benthic phosphorus cycle, contributed to increased marine productivity and carbon burial over an extended period of time during OAEs.

# 1 Introduction

The Cretaceous period was characterized by high atmospheric $CO_2$ levels and temperatures and by episodic deposition of
35 sediments with extremely high organic carbon content, referred to as Oceanic Anoxic Events (OAEs; e.g., Schlanger and Jenkyns, 1976; Jenkyns, 1980). These events represent prolonged and intense perturbations of the global carbon cycle (e.g., Arthur et al., 1985) associated with widespread anoxia in most ocean basins. Oceanic Anoxic Events are also characterized by positive carbon isotope ($\delta^{13}$C) excursions that have been related to globally enhanced rates of organic carbon burial (e.g., Berger and Vincent, 1986; Kump, 1991). The most prominent of these events were OAE1a during the early Aptian and
40 OAE2 at the Cenomanian/Turonian boundary. Smaller-scale events of probably more regional extent were also identified in several ocean basins (e.g., OAE 1b, c, d and OAE3). Another such event, the mid-Cenomanian event (MCE; Coccioni and Galeotti, 2003), appears to represent a less intense precursor event of OAE2. However, detailed records of the MCE are still sparse with most studies focusing on the higher amplitude events (OAE1a, b, c, d and OAE2). Records of the MCE until now are predominantly from the North Atlantic and Tethys region (e.g., Umbria Marche Basin (Coccioni and Galeotti,
2003); English Chalk (Gale, 1989; Jenkyns et al., 1994); Western Interior Seaway (Keller et al., 2004); Blake Nose (Ando et al., 2009), displaying a positive isotope excursion during the *Thalmanninella reicheli* foraminiferal zone. In shelf areas of the global ocean, major sea level changes associated with the cycles Ce2.1 and Ce3 of Gale et al. (2002) may have caused long lasting hiatuses obliterating evidence of the MCE.

The triggering mechanisms and internal processes that were essential for sustaining high primary productivity over extended
periods of time during OAEs remain enigmatic. One of the limitations for understanding the driving mechanisms and dynamics of OAEs is the uncertainty about the duration of OAEs and individual phases within these events. Previous estimates vary substantially (Table 1), partly due to differing definitions of the onset and end of OAEs, based on the extent of organic-rich sediment accumulation or the $\delta^{13}$C excursion. For instance, the positive shift in the initial part of the excursion is considered to represent the entire OAE1a (e.g., Li et al, 2008; Moullade et al., 2015), whereas the return to
background values (e.g., Sageman et al., 2006) or the end of the $\delta^{13}$C plateau (e.g., Eldrett et al., 2015) are included into the definition of OAE2.

Carbon isotope excursions associated with OAEs often display an initial negative excursion, which has been attributed to the rapid release of a large volume of $^{13}$C-depleted carbon either as methane and $CO_2$ from organic material in terrigenous soils and sediments, dissociation of submarine methane hydrates, or direct volcanic exhalation of $CO_2$ and thermal combustion of
60 organic-rich sediments driven by volcanic heat flux (e.g., Dickens et al., 1995; Jenkyns, 2003; Erba, 2004; Turgeon and Creaser, 2008; Du Vivier et al., 2014). The ensuing positive $\delta^{13}$C excursion is generally attributed to enhanced burial rates of $^{12}$C enriched organic carbon in marine organic-rich shales and/or in terrestrial peat and coal deposits (e.g., Scholle and Arthur, 1980; Jenkyns, 1980; Schlanger et al., 1987; Arthur et al., 1988). Key constraints on the subsequent feedback mechanisms of nutrient and carbon cycles are the rates at which the carbon was released and buried. Fast release rates are
65 consistent with catastrophic events, such as methane hydrate dissociation or thermal combustion of organic-rich sediments,

whereas slower rates implicate other processes. Burial rates influenced the balance between continuing release of [13]C-depleted carbon and the relative impact of sequestration through enhanced biological productivity and globally intensified and expanded ocean anoxia.

Different hypotheses have been put forward to explain the enhanced accumulation of organic matter (e.g., Arthur et al., 1988; Jenkyns, 2010) and the processes triggering and maintaining global anoxia and enhanced primary productivity during Cretaceous OAEs. These include fertilization by nutrient input into the ocean system in association with the activity of large igneous provinces (LIPs) (e.g., Schlanger et al., 1981; Larson, 1991; Trabucho-Alexandre et al., 2010), sea level controlled remobilization of nutrients from flooded low altitude land areas associated with major marine transgressions (e.g., Jenkyns, 1980; Mort et al., 2008), increased phosphorus input resulting from intensified weathering on land (e.g., Larson and Erba, 1999; Poulton et al., 2015), or release of phosphorus as a main limiting nutrient from sediments into the water column under anoxic bottom water conditions (e.g., Ingall and Jahnke, 1994; Slomp and Van Cappellen, 2007). However, the extent and the conditions under which phosphorus was available to act as a fertilizer for marine productivity as well as the mechanisms and internal feedbacks that sustained OAEs for several hundred thousand years remain controversial.

Primary production in the modern marine environment is mainly limited by the availability of nitrogen, iron and phosphorus with the latter considered the ultimate limiting nutrient on longer geological timescales (Holland, 1978; Broecker and Peng, 1982; Smith, 1984; Codispoti, 1989; Tyrell, 1999; Filipelli, 2008). It has been suggested that the phosphorus budget of the modern ocean is imbalanced, since input fluxes from riverine, aeolian and ice-rafted sources do not fully match phosphorus burial in marine sediments and the hydrothermal removal of dissolved phosphate from the deep ocean (e.g., Wallmann, 2010). The main source for phosphorus in the oceans is the terrigenous discharge from rivers in the form of dissolved inorganic phosphorus (DIP) and organic phosphorus (DOP), particulate inorganic phosphorus (PIP) and particulate organic phosphorus (POP). The delivery rate of POP and immediately bioavailable, reactive DIP and DOP is mainly controlled by continental weathering and seasonal riverine discharge (Ruttenberg, 2003; Li et al., 2017a) and, thus, closely linked to variations in atmospheric carbon dioxide concentrations and the hydrological cycle. Most sedimentary rocks and sea floor sediments are characterized by low concentrations of phosphorus <0.13 % (Riggs, 1979) and shallow marine environments are considered to be the main phosphorus sinks (Ruttenberg, 1993). Exposure to well-oxygenated water masses on the shelf and slope leads to almost complete remineralisation of organic matter and precipitation of authigenic phosphorus minerals (Ruttenberg, 1993).

In deep-sea sediments underlying well-oxygenated bottom water masses, phosphorus remains mainly bound to manganese- and iron-oxides and -hydroxides and as authigenic calcium-bound phosphorus (Supplementary Material S1), which typically exhibits C:P below the Redfield ratio. Excess phosphate is released from shelf and continental margin sediments deposited in low oxygen environments, resulting in elevated sedimentary C:P. Today, estimates of residence time vary between 10 and 17 kyr (Ruttenberg, 2003) and 80 kyr (Broecker and Peng, 1982), depending on estimated burial rates within the different marine phosphorus sinks, in particular in shallow seas and along continental margins (Ruttenberg, 2003). During Cretaceous

OAEs, the extent and burial efficiency of these sinks must have varied substantially, due to sea-level and redox oscillations affecting marginal seas (e.g., Danzelle et al., 2018).

There are conflicting views on the influence of expanded Oxygen Minimum Zones (OMZs) or oceanic anoxia on the phosphorus cycle and, in particular, whether OMZ sediments serve as a phosphorus source or sink. It has been argued that oxygen depleted bottom waters favour phosphorus release from the sediment to the water column (Ingall and Jahnke, 1994) and stimulate primary production in surface waters (Wallmann, 2003). This in turn results in increased organic carbon export flux, leading to higher oxygen demand, OMZ expansion and intensification and a positive feedback with benthic phosphorus release (Slomp and Van Cappellen, 2007; Wallmann, 2010). By contrast, other studies suggested that intensified phosphorus burial occurs under anoxic conditions in shallow water environments, based on observations of calcium fluorapatite (CFA) precipitation in present-day shallow water oxygen-depleted upwelling areas (Schulz and Schulz, 2005; Arning et al., 2009a, b; Goldhammer et al., 2010; Ingall, 2010; Cosmidis et al., 2013). On the long-term, the formation of CFA is approximately in balance with enhanced phosphorus release from anoxic sediments, implying that the dissolved oceanic phosphorus inventory is largely unaffected by regional changes in oxygen concentrations (Delaney, 1998; Anderson et al., 2001; Roth et al., 2014). However, this equilibrium may have been disturbed during periods of deep-water anoxia, as in the Mediterranean Sea during sapropel formation (Slomp et al., 2004) and in the Cretaceous ocean during OAEs (e.g., Mort et al., 2007, 2008). Observations and model simulations indicated that global warming enhances the terrestrial input of biologically reactive phosphorus to the marine environment, leading to increased production and burial of organic carbon in marine sediments (Mackenzie et al., 2002).

In this study, we focus on the two most intense OAEs, OAE1a and OAE2, which occurred during intervals of extreme greenhouse gas forcing within the Cretaceous period. We analyse extended, continuous sediment successions from two drill cores in the Vocontian Basin (southern France) and Tarfaya Basin (southern Morocco) (Fig. 1). Time series analyses of high-resolution X-Ray fluorescence (XRF) scanning elemental and natural gamma-ray (NGR) borehole logging data provide constraints on the duration of individual phases within these carbon isotope excursions and new insights into the response of the ocean-climate system to these extreme carbon cycle perturbations. In addition, we analyse phosphorus speciation across the mid-Cenomanian carbon isotope event (MCE) and OAE2 to investigate relationships to changing sea level, OMZ intensity and carbon burial and to test the hypothesis of redox controlled phosphorus release as a nutrient source initiating and/or enhancing carbon burial during OAEs.

## 2 Material and Methods

### 2.1 Sediment cores

Core SN°4 (27°59'46.4'' N, 12°32'40.6'' W) was retrieved in the Tarfaya Basin (southern Morocco), 40 km east of the town Tarfaya, close to the road to Tan Tan. The 350 m long marine sediment succession in Core SN°4, which provides an expanded record of OAE2, was deposited in an outer shelf setting within a subsiding basin on a passive margin during the

Late Cretaceous. Late Albian to Turonian stable isotope and geochemical records were previously presented by Beil et al. (2018) and Scholz et al. (2019). High-resolution XRF-scanner, bulk sediment stable isotope and carbon records across the onset of OAE2 were provided by Kuhnt et al. (2017). Here, we complement these records with phosphorus speciation data and time-series analysis of logging natural gamma ray and high resolution XRF-scanner elemental distribution data across OAE2.

Cores LB1 (43°14'37'' N, 5°34'12'' E; 70 m long) and LB3 (43°14'42'' N, 5°34'52'' E; 56 m long) were drilled near Roquefort-La Bédoule, 16 km southeast of Marseille in southern France. During the Aptian, the coring sites were located within an isolated intrashelf basin, the South Provence Basin, on the North Provençal carbonate platform, where an expanded succession of marine sediments including OAE1a accumulated. Details of the drilling operation were published by Flögel et al. (2010), lithologic descriptions, biostratigraphy and intermediate-resolution carbonate and stable isotope measurements were provided by Lorenzen et al. (2013) and Moullade et al. (2015). Here, we present new high-resolution bulk carbonate isotope data across the precursor, onset and peak phases of OAE1a as well as time-series analysis of logging natural gamma ray and XRF-scanner elemental distribution data from a spliced composite record of LB1 and LB3. The tie point between both cores is at a depth of 49.4 m (Section 34, 3 cm section depth) in LB3 and 10.83 m (Section 8, 95 cm section depth) in LB1, below bed 170, identified by Moullade et al. (2015) as a prominent correlating feature. The tie point was selected at the base of a prominent, extended maximum in XRF-scanner derived Log(Terr/Ca) (Section 2.3), identified in both cores.

## 2.2 NGR-logging

Wireline borehole logging of the SN°4 drill hole was carried out by Geoatlas (Laayoune, Morocco) using a Century geophysical logging system with natural gamma ray (NGR) sensor. Detailed information on NGR logging of SN°4 is given in Beil et al. (2018). For borehole logging of the LB1 and LB3 drill holes, an Antares Aladdin logging system with GR5 sensor probe was deployed. All logging operations were conducted shortly after completion of the drilling operation in boreholes without metal casing and NGR data are presented as American Petroleum Institute radioactivity units in counts per second (cps) with vertical resolutions of 0.1 m for SN°4, 0.05 m for LB1 and 0.025 m for LB3. The intensity of natural gamma radiation is predominantly influenced by the concentration of three different elements: Potassium (K), Uranium (U) and Thorium (Th). All three elements are bound to clay minerals with uranium also adhesively enriched in organic matter. In environments with low terrigenous input and high organic matter accumulation as in Core SN°4, NGR is predominantly controlled by the concentration of organic matter. By contrast, clay accumulation mainly controlled NGR in Cores LB3/LB1, where the organic matter content of the sediment is low.

## 2.3 XRF-scanning and line scan imaging

Detailed descriptions of XRF-scanning of Core SN°4 are provided in Kuhnt et al. (2017) and Beil et al. (2018). Sections were scanned with a second generation Avaatech X-ray fluorescence scanner at the Christian-Albrechts-University, Kiel. Surfaces were covered with a 4 µm thick Ultralene foil after cleaning with fine-grained sandpaper. Data for this study are

from the 10 kV dataset scanned with a spatial resolution of 1 cm (vertical slit of 1 cm, horizontal slit of 1.2 cm) measured with 10 kV, 750 μA, no filter and 10 s acquisition time. Sections of Cores LB1 and LB3 were polished with fine-grained sandpaper and covered with 4 μm thick Ultralene foil prior to scanning with the Avaatech XRF scanner at the Christian-Albrechts-University, Kiel. The XRF-data from the 10 kV dataset presented in this study were measured with a spatial resolution of 1 cm with the tube setting of 10 kV, 250 μA and no filter with 15 s acquisition time.

Raw XRF spectra of all three cores were converted with the iterative least-square package *Win Axil batch* (Canberra Eurisys) and the 10 kV Kiel model into element specific area counts. The Jai CV-L 107 3 CCD color line scan camera installed in the Avaatech XRF-scanner at the Christian-Albrechts-University, Kiel, was used to obtain core images with a downcore resolution of 143 ppcm for Cores SN°4, LB1 and LB3. We used the logarithmic ratios of elemental counts to eliminate XRF-scanning specific effects such as matrix-effect, grain size effect or variations in rock density (Weltje and Tjallingii, 2008). Log(Terr/Ca) is used as a proxy for terrigenous, clastic material vs. marine biogenic carbonate. Elements of typically terrigenous origin (Terr) detected with the 10 kV settings are aluminium (Al), iron (Fe), potassium (K), silicon (Si) and titanium (Ti) (e.g., Peterson et al., 2000; Calvert and Pedersen, 2007; Mulitza et al., 2008; Tisserand et al., 2009; Govin et al., 2012). The variability of Fe closely matches the variability of the other terrigenous elements (Supplementary Material S2), thus appearing not to be influenced by changing redox conditions. Calcium (Ca) is of marine origin, mainly from calcareous nanno- and microplankton. Log(K/Al) is widely used to characterize the composition of clay mineral assemblages (Weaver, 1967, 1989; Niebuhr, 2005). The ratio is primarily controlled by variations in the amount of the potassium-rich illite and thus reflects the intensity of physical/chemical weathering in the source area (Calvert and Pedersen, 2007).

## 2.4 Stable Isotopes

Organic and bulk carbonate stable isotope data from Core SN°4 were compiled from the high-resolution record over the onset of OAE2 (Kuhnt et al., 2017) and from the lower resolution record over the entire core (Beil et al., 2018). Bulk carbonate samples were analysed with the Finnigan MAT 251 and MAT 253 mass spectrometers at the Leibniz Laboratory for Radiometric Dating and Stable Isotope Research at the Christian-Albrechts-University, Kiel. The accuracy for carbon isotopes was ±0.05 ‰ and ±0.08 ‰ for oxygen isotopes. Organic carbon stable isotopes were analysed at the GeoZentrum Nordbayern (Erlangen) with a Flash EA 2000 elemental analyzer coupled to a Thermo Finnigan Delta V Plus mass spectrometer on samples with a spacing between ~0.1 and ~2.3 m. Measurement accuracy was ±0.06 ‰. Results are reported on the delta scale as $\delta^{18}O$, $\delta^{13}C_{carbonate}$ and $\delta^{13}C_{org}$ against the Vienna PeeDee belemnite standard (VPDB).

Initial stable isotope data from Cores LB1 and LB3 were presented in Lorenzen et al. (2013) and Moullade et al. (2015). Published data have a spacing of ~20 cm in LB1 and ~40 cm in LB3. New samples were analysed to increase the resolution to ~5 cm during the early part of OAE1a between 18 and 41.5 m in Core LB1. Stable isotopes of bulk carbonate samples were analysed at the Leibniz Laboratory for Radiometric Dating and Stable Isotope Research at the Christian-Albrechts-University, Kiel, with a Finnigan MAT 251 mass spectrometer. Analytical uncertainty for stable carbon isotopes was ±0.04

‰ and ±0.07 ‰ for stable oxygen isotopes. Results are reported on the delta scale as $\delta^{18}O$ and $\delta^{13}C_{carbonate}$ against the Vienna PeeDee belemnite standard (VPDB).

## 2.5 Total organic carbon, carbonate and nitrogen content

Aliquots of samples for determination of major and trace elements and phosphorus speciation in Core SN°4 were analysed for total organic carbon (TOC), carbonate and nitrogen content. Replicates of 8-10 mg sample material were pulverized, homogenized and sealed into tin capsules for the determination of Total Carbon (TC) and Nitrogen (N) and into silver capsules followed by decarbonatization with 0.25 N hydrochlorid acid (HCl) to measure TOC. Different weights of Acetanilid (10.36 % N, 71.09 % C) and of the certified soil standard BSTD1 (0.216 % N, 3.5 % C) were sealed and measured for calibration and for monitoring of long-term stability. All capsules were measured with a Carlo Erba Proteinanalyzer NA1500 at the Geomar Helmholtz Centre for Ocean Research, Kiel. Carbonate content (TIC) was calculated as TIC = (TC – TOC) * 8.3333.

## 2.6 Time series analysis

The package *astrochron* (Meyers and Sageman, 2007; Meyers et al., 2012a; Meyers, 2014) for R (R Core Team, 2017) was used to perform frequency analysis on the NGR and XRF Log(Terr/Ca) datasets. The NGR datasets were linearly interpolated to a spatial resolution of 0.1 m for Core SN°4 and 0.025 m for the spliced dataset of Cores LB1 and LB3. XRF-scanner derived Log(Terr/Ca) from LB3/LB1 was linearly interpolated to a spatial resolution of 0.01 m. Long-term variability of >20 m wavelengths was removed from the NGR dataset by using a Gaussian kernel smoother with the function *noKernel* of *astrochron*.

Dominant frequencies in Cores LB1 and LB3 were extracted with the robust red noise MTM analysis of Mann and Lees (1996) using the function *mtmML96* (Patterson et al., 2014) of *astrochron* with five 2π prolate tapers. Visualization and checks were performed with the *eha*-function of *astrochron* with three 2π prolate tapers and a window of 6 m over the frequency range between 0 and 4 cycles/m for the spliced record of LB1 and LB3. Identical intervals and frequency ranges were used to check results with the complementary XRF-scanner derived Log(Terr/Ca) (Supplementary Material S3.1).

An age model for Core SN°4 was derived by correlating the NGR record to that of the neighbouring Core S13 (Meyers et al., 2012a) following Kuhnt et al. (2017). A total of 14 tiepoints delimiting characteristic features of both cores were used (Supplementary Material S4). The age model of Meyers et al. (2012a) was transferred into chronological ages by anchoring the Cenomanian/Turonian boundary at the top of cycle 3 to the chronological age of 93.9 Ma (Meyers et al., 2012b).

Periodicities for precession, obliquity and eccentricity are based on the orbital solution La04 (Laskar et al., 2004). Additional periodicities for eccentricity were extracted from the orbital solutions La10a-d (Laskar et al., 2011a) for the Cenomanian and Aptian intervals and La10a-d and La11 (Laskar et al. 2011a, b) for the Cenomanian interval using the *astrochron*-function *mtmML96* (Patterson et al., 2014). The analysis of orbital parameters for OAE1a was limited to the Aptian interval between

113 and 126 Ma and to the Cenomanian interval between 93 and 99 Ma for OAE2 (Supplementary Material S5). Ratios were calculated for the main frequencies. Periodicities and ratios of the different orbital cycles are presented in Supplementary Material S5.1 and S5.2 for the Aptian and in Supplementary Material S5.3 for the Cenomanian interval.

## 2.7 Phosphorus speciation and analysis of major and trace elements

The distribution of major and trace elements and phosphorus speciation were determined on new samples from Core SN°4, mainly taken close to the published organic stable isotope samples (Beil et al., 2018). Samples cover the interval between 42.11 m (Section 18, Segment 3, 23-24 cm) and 305.77 m (Section 111, Segment 2, 29-31 cm). The core was sampled every ~2.4 m with increased resolution of ~1.2 m over the global isotope excursions within OAE2 and the MCE. The surface of samples was cleaned with a metal-free porcelain knife and ground down with a rotary mill with agate balls to prevent

metallic contamination. The powdered material was subdivided into aliquots for major and trace elements analysis and material for determination of phosphorus speciation. A further aliquot was used for carbonate and organic carbon measurements for samples not close to the already published organic isotope samples. Additional samples from high carbonate content intervals were prepared and measured to determine the influence of low clay and organic content.

### 2.7.1 Analysis of major and trace elements

Aliquots of 100 mg were weighed into PTFE vessels. Each sample was treated with 2 ml HF, 2 ml $HNO_3$ and 3 ml $HClO_4$, sealed and heated for 8 hours at 185 °C. The acid was subsequently smoked off at 190 °C. The almost dry residue was dissolved again in 1 ml $HNO_3$, smoked off and dissolved again in 5 ml ultrapure water and 1 ml $HNO_3$ to be heated again for 2 hours in sealed vessels. The solution was finally transferred and diluted with 10 ml $HNO_3$ in volumetric flasks. The solutions were measured with a VARIAN 720-ES ICP-OES for major elements and an Agilent Technologies 7500 Series

ICP-MS for trace elements at GEOMAR Helmholtz Centre for Ocean Research Kiel. Accuracy for phosphorus concentrations based on repeated measurements of the standard MESS-3 (National Research Council of Canada) was 2.2 % (n=15). Precision based on duplicate measurements of samples was on average 0.9 % (n=15).

### 2.7.2 Phosphorus speciation

Samples for the measurement of phosphorus speciation were aliquots from samples used for the analysis of major and trace

elements. A modified CONVEX extraction method (Oxmann et al., 2008; Supplementary Material S6) was used to extract Ca-bound and Al/Fe-oxyhydroxide bound phosphorus. The sample material (250 mg) was subsequently treated with 3.75 ml KCl/EtOH, decanted and supernatants discarded three times. Al/Fe-bound phosphorus was extracted from the centrifuged residue with the addition of first 3.75 ml $NaOH/Na_2SO_4$ (incubated for 1 h at 25 °C), then 3.75 ml $NaOH/Na_2SO_4$ (incubated for 2h at 99 °C) and finally 7 ml $Na_2SO_4$. Solutions were centrifuged and decanted after each of the three treatment steps and

stored for measurement. The centrifuged residue was decarbonated for 8 h with 3.75 ml $H_2SO_4$. The Ca-bound phosphorus fraction was extracted by adding further 3.75 ml $H_2SO_4$ (incubated for 2 h at 99 °C) and twice $Na_2SO_4$ (3.75 and 7 ml). Each

of the three steps was followed by centrifugation and decanting of the solution. Supernatants of each step were again mixed and stored for measurements. Details are provided in Supplementary Material S6.1.

For measurement of Ca-bound phosphorus, an aliquot of 0.1 ml was diluted with 4.9 ml MilliQ. Concentrations of Al-/Fe-bound phosphorus were measured on 4.5 ml of sample solution equilibrated with the addition of 0.5 ml of sol7 (Supplementary Material S6.2) to pH of ~1. Concentrations of Al/Fe- and Ca-bound phosphorus were calibrated with the $PO_4$-Merck Standard (1000 mg $PO_4$/l) diluted to specific concentrations. The ammonium molybdate solution (0.1 ml; Supplementary Material S6.3) was added 30 min before measurements. For samples close to the photometric saturation level a new aliquot was diluted to lower concentration within the measurement range. A detailed list with used chemicals and respective concentrations is available in Supplementary Material S6.2.

The atomic ratio of organic carbon to $P_{reactive}$ (defined as the sum of loosely absorbed, organic, authigenic and iron bound phosphorus) was interpreted by Anderson et al. (2001) as a proxy for marine paleoproductivity. Due to sink switching (transfer to different phosphorus pools) between the different phosphorus species. especially during early diagenesis, the authors assumed that phosphorus in the different pools was originally derived from the remineralization of organic material and could therefore be considered equivalent to $P_{org}$. Here, $P_{react}$ is calculated by summing up the two CONVEX-derived P-pools CaP (seen as equivalent to $P_{auth}$) and AlFeP. Both loosely absorbed and organic P were not measured. It is assumed that the concept of Andersen et al. (2001) is applicable, as both extracted P-pools (CaP and AlFeP) contain the majority of $P_{total}$ (mean = 89 %; Supplementary Material S7.1).

### 2.7.3 Phosphorus and organic carbon accumulation rates in Core SN°4

Bulk sediment mass accumulation rates (MAR) were calculated by multiplication of linear sedimentation rates (LSR; derived from the age model of Core SN°4) with an average dry bulk density (DBD) of 2.1 g cm$^{-3}$. Mean density for the interval encompassing OAE2 (600-2000 kyr) in neighbouring Core S13 (Meyers et al., 2012a) is 2.08 g cm$^{-3}$ (StDev 0.12 g cm$^{-3}$). Phosphorus accumulation rates (PAR) were calculated by multiplying MAR with phosphorus concentrations from discrete measurements. Organic carbon accumulation rates (TOCAR) were calculated using the organic carbon concentrations from Kuhnt et al. (2017) and Beil et al. (2018).

### 3 Results

### 3.1 Temporal evolution of bulk carbonate and organic carbon δ$^{13}$C across OAE1a and OAE2

The high-resolution δ$^{13}$C record of OAE1a in Core LB3/LB1 allowed correlation to the Aptian isotope stages C2 to C8 proposed by Menegatti et al. (1998). We also identified prominent carbon isotope maxima a-c (following Voigt et al., 2007) in the SN°4 record of OAE2. We determined five phases (precursor, onset, peak, plateau and recovery) in the δ$^{13}$C records of both OAE1a and OAE2 (Figs. 2, 3 and Supplementary Material S8). The interval preceding both OAEs is characterized by a

long-term mean without discernible trends and low variability. A precursor phase of two to four distinct $\delta^{13}$C minima (negative excursions) precedes the onset of the positive carbon isotope excursion (segment C3 of the Aptian carbon isotope curve of Menegatti et al., 1998). The onset phase encompasses the entire interval of increasing $\delta^{13}$C towards the first prominent peak (peak a of OAE2 following Voigt et al., 2007). This interval corresponds to segment C4 of the Aptian carbon isotope curve of Menegatti et al., 1998. The peak phase is defined as the interval starting with maximal values (first peak) to the end of the second peak (peak b of OAE2 following Voigt et al., 2007). A characteristic plateau phase (interval of relatively constant $\delta^{13}$C) follows the second peak of the carbon isotope excursions (C6 and lower part of C7 of Menegatti et al., 1998). The recovery phase encompasses the return of $\delta^{13}$C-values to background levels following the OAE excursions (upper part of C7 and C8 of Menegatti et al., 1998).

The $\delta^{13}$C decreases during the precursor phase have comparable amplitudes with 0.7 ‰ for OAE1a and 0.4 ‰ for OAE2 (Figs. 2 and 3). The most positive values of the OAE2 $\delta^{13}$C isotope excursion are reached at the first peak (peak a following Voigt et al., 2007) at 103.22 m with -24.2 ‰. By contrast, values continue to increase during OAE1a with highest values of 4.9 ‰ during the plateau stage at 40.11 m. The difference between the last minimum of the precursor phase and the first maximum (peak a for OAE2) of the peak phase is 2.4 ‰ for the carbonate $\delta^{13}$C record of OAE1a and 4.6 ‰ for the organic carbon $\delta^{13}$C record of OAE2. Considering the differing trend of OAE1a, with further increases during the plateau stage, the total amplitude of OAE1a is 3.6 ‰. Differences in mean values between the background levels before and after the OAE are 0.3 ‰ for OAE2 (from -28.5 ‰ (StDev 0.3 ‰) before to -28.2 ‰ (StDev 0.2 ‰) after OAE2) and 0.8 ‰ for OAE1a (from 1.9 ‰ (StDev 0.1 ‰) before to 2.7 ‰ (StDev 0.2 ‰) after OAE1a) indicating a general increase in background $\delta^{13}$C after the OAEs.

## 3.2 Temporal evolution of bulk carbonate $\delta^{18}$O across OAE1a and OAE2

The $\delta^{18}$O curves share common trends, except for the cyclic lithological changes in the upper part of the sedimentary record of OAE1a in LB3. Transient cooling events, identified by $\delta^{18}$O increases in LB3/LB1 and SN°4, occur during the early phases of both OAE1a and OAE2 (Figs. 2, 3 and Supplementary Material S9). A first prominent cooling event prior to the onset of OAE2 in Core SN°4 (Kuhnt et al., 2017), which is not identified at other localities, was probably associated with local upwelling of cooler deep water masses in the Tarfaya Basin. Cooling during OAE2 occurred in three main steps starting within the onset phase of the positive carbon isotope excursion. The most intense cooling, associated with the Plenus Cold Event, occurred during the peak phase of the excursion (in the trough between the $\delta^{13}$C peaks a and b). The Plenus Cold Event is globally recorded (e.g., Forster et al., 2007; Sinninghe Damsté et al., 2010; Jarvis et al., 2011; Jenkyns et al., 2017) and coincided with invasion of boreal species in the European Chalk Sea (Gale and Christensen, 1996; Voigt et al., 2003), extinction of the planktic foraminifer *Rotalipora cushmani* (e.g., Kuhnt et al., 2017) and re-oxygenation of bottom water masses (e.g., Eicher and Worstell, 1970; Kuhnt et al., 2005; Friedrich et al., 2006). OAE1a shows a similar response of global temperatures to enhanced organic carbon burial (Kuhnt et al., 2011; Jenkyns, 2018): the main $\delta^{18}$O increase during

the latter part of segment C4 in the $\delta^{13}C$ curve of Meneghatti et al. (1998) also occurs during the onset phase. A further

cooling event within segment C6 follows transient warming during the peak phase. Jenkyns (2018) recognized these transient coolings, also recorded in the northeastern Atlantic Ocean (Naafs and Pancost, 2016), Italy (Bottini et al., 2015), Turkey (Hu et al., 2012) and in the Pacific Ocean (Dumitrescu et al., 2006), as global events. Similarities in the $\delta^{18}O$ records across both OAEs imply a similar response of the ocean-climate system to lowered atmospheric $p$CO$_2$ levels due to excess carbon drawdown associated with burial of vast amount of organic material on a global scale.

**3.3 Regional differences in terrigenous input during OAE1a and OAE2**

The different paleogeographic settings and depositional environments of the Tarfaya Basin (Core SN°4) and South Provence Basin (Cores LB1 and LB3) result in important differences in terrigenous sediment input. The XRF-derived Log(Terr/Ca) (Fig. 2) in the Tarfaya Basin exhibits low variability throughout with lowest values during OAE2. By contrast, Log(Terr/Ca) (Fig. 3) shows higher amplitude variability during OAE1a. A major increase in Log(Terr/Ca) from -1.69 to -0.87 during the

330 onset of OAE1a (68.09-67.99 m) indicates either a decrease in carbonate deposition or an increase in terrigenous input. Following this increase, the NGR and Log(Terr/Ca) records exhibit high amplitude and high frequency variability during C4 to C6 (26-23.29 m) and during the latter part of C7 and C8 (49.7-0 m) (Figs. 3, 4 and Supplementary Material S3.1).

**3.4 Duration of OAE1a estimated from time series analysis of NGR and XRF-scanner data**

The spliced NGR record of LB1 and LB3 across OAE1a was subdivided into five intervals of relatively consistent orbital

periodicities (N1 to N5), based on EHA-analysis (Fig. 4, Supplementary Material S3). The recognition of cyclic patterns in the NGR record during OAE1a allowed correlation of major frequencies with orbital periodicities (Laskar et al., 2004, 2011a, 2011b) and calculation of mean sedimentation rates (Fig. 4, Supplementary Material S3.2 S3.3 and 5.2). Comparison with durations proposed by Malinverno et al. (2010) and Scott (2016) for the different C-stages of Menegatti et al. (1998) is shown in Table 2. A strong response to variations in obliquity and precession is evident in the latest part of stage C2, which

precedes the onset of OAE1a (Fig. 4, Supplementary Material S3.1 and S3.2). The NGR and Log(Terr/Ca) time series (Fig. 4 and Supplementary Material S3.1) also exhibit precession-paced variations during the middle and later part of stage C7 and C8, which persist after OAE1a. This trend corresponds to pronounced lithological changes between carbonate-rich and clay-rich beds that are also apparent in the core images and the $\delta^{13}C_{carbonate}$ and $\delta^{18}O$ profiles (Fig. 3).

Sedimentation rates (Supplementary Material S3.3) increase from C3 and C4 (1.4 cm kyr$^{-1}$) to the recovery phase C8 (5.1 cm

345 kyr$^{-1}$). The relatively stable ratio between terrigenous and carbonate content (Log(Terr/Ca) suggests a continuous increase in both carbonate and terrigenous input (Fig. 3).

### 3.5 Duration of OAE2 estimated from time series analysis of NGR data

EHA-analysis of NGR data reveals a marked response to obliquity (mainly o1, 48 kyr) and short eccentricity (e2/e3, 100 kyr) during the positive $\delta^{13}C$ excursion (Fig. 5). Following this interpretation, the $\delta^{13}C$ excursion of OAE2 in Core SN°4 lasted for ~790 kyr including the recovery phase (~360 kyr without the recovery phase). The response to orbital forcing was more pronounced during the peak phase (~90 kyr) and the recovery phase (~440 kyr) than during the plateau phase (~200 kyr). Sedimentation rates before the onset of OAE2 were 8.2 cm kyr$^{-1}$, dropping to 4.1 cm kyr$^{-1}$ during the positive $\delta^{13}C$ excursion and recovering to ~8 cm kyr$^{-1}$ in the upper part of the plateau phase (Supplementary Material S4.1), which is comparable to levels prior to OAE2.

Four $\delta^{13}C_{org}$ minima (precursor phase) occur over a period of ~130 kyr prior to the positive $\delta^{13}C$ excursion of OAE2 (Fig. 2, Supplementary Material S4.1). The onset phase of OAE2 (~70 kyr), defined as the interval between the centre of the last $\delta^{13}C_{org}$ minimum and the first $\delta^{13}C_{org}$ maximum of OAE2, is subdivided into three phases: the initial steep increase lasted ~31 kyr, the intermediate plateau ~14 kyr and the final rise to the first peak (peak a of Voigt et al., 2007) of the carbon isotope excursion ~24 kyr. The time interval between the two maxima in $\delta^{13}C_{org}$ (peak phase) was ~90 kyr. The duration of the plateau phase defined as the period between the second maximum and the end of the $\delta^{13}C_{org}$ plateau was ~200 kyr. The recovery phase between the end of the plateau and the return to background values after OAE2 lasted ~440 kyr. A comparison with durations proposed by Li et al. (2017b) and Gangl et al. (2019) is provided in Table 3.

### 3.6 Comparison of the durations of OAE1a and OAE2

The precursor phase of OAE1a (C3 of Menegatti et al., 1998), which consists of an extended interval of low $\delta^{13}C_{carbonate}$ (<1.75 ‰) and encompasses two prominent $\delta^{13}C$ minima (<1.5 ‰), lasted ~434 kyr. The precursor phase of OAE2 is characterized by four nearly equally spaced $\delta^{13}C_{org}$ minima (<28.5 ‰) and lasted ~126 kyr. The last two prominent minima immediately precede the onset of OAE2 and lasted for ~75 kyr. The onset phase of the positive isotope excursions had a duration of ~388 kyr for OAE1a (C4 of Menegatti et al., 1998) and ~68 kyr for OAE2. The peak phase lasted ~596 kyr in OAE1a (C5 and C6) and ~86 kyr in OAE2, the plateau phase extended over ~1343 kyr in OAE1a (C7) and ~204 kyr in OAE2 and the final recovery phase had a duration of ~377 kyr for OAE1a (C8) and ~435 kyr for OAE2. The precursor, onset and peak phases were consistently ~5 times longer in OAE1a than in OAE2, whereas the plateau phase was more than ~7 times longer. The recovery phases had the same duration for both OAEs, taking into account uncertainties in defining change points at the end of the $\delta^{13}C$ excursion.

### 3.7 Sedimentary phosphorus

Total phosphorus ($P_{total}$) concentrations in Core SN°4 (Fig. 6) are overall below 5 mg g$^{-1}$ except during two periods of enrichment peaking at 220.19 m with 16.61 mg g$^{-1}$ and at 104.4 m with 7.53 mg g$^{-1}$. Corresponding peaks in $P_{total}$/Al give a weight ratio of 1.07 at 220.19 m and 1.20 at 104.4 m. Concentrations and variability of $P_{total}$ are low in the lowermost (305-

267 m) and uppermost (67.51-42.11 m) parts of the studied interval with means of 0.54 mg g$^{-1}$ (StDev 0.19 mg g$^{-1}$) and 0.86 mg g$^{-1}$ (StDev 0.3 mg g$^{-1}$), respectively. The origin of the phosphorus is difficult to assess, as the different species extracted are defined by the CONVEX method (Oxmann et al., 2008). Earlier studies found finely distributed fish debris and fecal pellets in Cenomanian sediments of the Tarfaya Basin (e.g., El Albani et al., 1999) partially reprecipitated as phosphate nodules during early diagenesis (e.g., Leine, 1986; Kuhnt et al., 1997). Phosphatic particles were not observed on the core surfaces and were not apparent in the XRF data of Core SN°4. They were only encountered as minor components in the residues of micropaleontological samples. The dissolution of phosphorus fixed in iron-aluminium crusts and the degradation of organic matter followed by reprecipitation as calcium bound phosphorus further complicate reconstruction of the initial source of phosphorus.

### 3.8 Phosphorus speciation

Concentrations of reactive phosphorus (P$_{react}$; Fig. 7) in Core SN°4, calculated by summing up Al/Fe- and Ca-bound phosphorus, are on average ~89 % of the total phosphorus (P$_{total}$; Figs. 6 and Supplementary Material S7.1) measured with ICP-OES. Organic-matter-bound phosphorus was not considered due to methodological limitations (Golterman, 2001) and to remineralization/reprecipitation (sink-switching) into more stable Al/Fe- and Ca-bound phosphorus species during early diagenesis. Differences between P$_{react}$ and P$_{total}$ are caused by not extractable phosphorus bound in insoluble minerals or adhesively bound phosphorus extracted and discarded during the first step with KCl/EtOH (Supplementary Material S6.1). Concentrations of phosphorus bound to aluminium- or iron-oxyhydroxides (AlFeP; Fig. 6) are low (median 0.014 mg g$^{-1}$). Increased concentrations (mean 0.027 mg g$^{-1}$) are determined for the lowermost interval between 305 and 272.58 m, followed by an interval with lower average concentration (mean 0.014 mg g$^{-1}$) and lower variability (StDev 0.006 mg g$^{-1}$) until 72.09 m. The uppermost interval is again characterized by increased variability (StDev 0.008 mg g$^{-1}$). Calcium-bound phosphorus (CaP; Fig. 6) is the dominant species in the studied interval of Core SN°4. Concentrations are below 4 mg g$^{-1}$ except for two periods of enrichment peaking at 220.19 and 104.4 m with 15.76 and 6.62 mg g$^{-1}$, respectively. Both maxima coincide with maximum enrichment of total phosphorus. The redox influence on Al/Fe-bound phosphorus in Core SN°4 is addressed in the Supplementary Material S10.

The atomic ratios of C$_{org}$ and N$_{total}$ against P$_{react}$ show very similar trends except for the lower part between 305.77 and 258.35 m that is characterized in N$_{total}$/P$_{react}$ by increased values (median 5.9) and in C$_{org}$/P$_{react}$ by low values and low variability (median 77.6, StDev 34.9), albeit in an interval with low resolution (Fig. 7). Afterwards both ratios show similar characteristics of low values (median of C$_{org}$/P$_{react}$ 89.1 and of N$_{total}$/P$_{react}$ 3.5) and low variability (StDev of C$_{org}$/P$_{react}$ 67.1 and of N$_{total}$/P$_{react}$ 1.9) until 104.4 m, punctuated by short-lived maxima during the MCE (average of C$_{org}$/P$_{react}$ 170 and of N$_{total}$/P$_{react}$ 7.5). OAE2 is again characterized by markedly increased ratios (maxima of C$_{org}$/P$_{react}$ 1123.8 and of N$_{total}$/P$_{react}$ 29.6) and the uppermost part exhibits increased values (median of C$_{org}$/P$_{react}$ 242.6 and of N$_{total}$/P$_{react}$ 7.3). The ratio of CONVEX-extracted AlFeP to P$_{total}$ (Fig. 7) shows increased values (mean 0.058) in the lower part between 305 and 272.58 m followed by low ratios (mean 0.014) to the top of the sampled interval in Core SN°4.

**3.9 Temporal changes in P-species concentrations and accumulation rates**

Two maxima in $P_{total}$ in the Cenomanian interval of Core SN°4 coincide with maxima in $P_{total}/Al$ (Fig. 6). The more prominent of these phosphorus enrichments (peaking at 220.19 m) precedes the onset of the MCE by ~3.04 m. The second maximum at 104.4 m occurs during the onset phase of OAE2 and coincides with a transient $\delta^{13}C$ minimum, approximately halfway towards the first $\delta^{13}C_{org}$ maximum at 103.22 m (Figs. 6 and Supplementary Material S9.1).

The ratio of Ca-bound to Al/Fe-bound phosphorus (Fig. 6) is characterized by three prominent peaks at 220.19, 143.51 and 104.40 m. The maxima at 220.19 and 104.40 m also exhibit high $P_{total}$ concentrations and $P_{total}/Al$ maxima. The first peak at 220.19 m precedes the first $\delta^{13}C_{org}$ increase of the MCE, and the second peak at 104.4 m occurs within the onset phase of the $\delta^{13}C_{org}$ excursion of OAE2. Both peaks of CaP/AlFeP and $P_{total}$ coincide with minima in the $C_{org}/P_{total}$-ratio (Supplementary Material S7.1), suggesting an inorganic source for phosphorus or enhanced recycling of organic carbon and reprecipitation of the organic-bound phosphorus as Ca-bound phosphorus.

The increase in the atomic $C_{org}/P_{total}$-ratio (Supplementary Material S7.1) during OAE2 postdates the increase in the atomic $C_{org}/N_{total}$-ratio, interpreted by Beil et al. (2018) as enhanced cycling of nitrogen rich organic matter within a dysoxic or anoxic water column. The increase in $C_{org}/P_{total}$ starts immediately above the prominent peak in Ca-/AlFe-bound and total phosphorus and coincides with the first $\delta^{13}C_{org}$ maximum of OAE2 (Figs. 6 and 7). The highest $C_{org}/P_{total}$ ratio is determined at 90.78 m within the plateau phase of OAE2. This peak coincides with minima in $P_{total}$ and $P_{total}/Al$, implying remobilization either synsedimentary due to preferential recycling of phosphorus in the water column and/or in the sediment. This decrease in $P_{total}$-content is paralleled by a decrease in $C_{org}$ (Fig. 6), suggesting enhanced remobilization of phosphorus as well as organic matter.

The atomic ratios of $C_{org}/P_{react}$ and $N_{total}/P_{react}$ (Fig. 7) are always lower or close to the Redfield ratio (C:N:P = 106:16:1, Redfield, 1958, 1963), except during the MCE and OAE2, when $C_{org}/P_{react}$ is equal to or higher than 106:1 and $N_{total}/P_{react}$ is close to the predicted ratio of 16:1. Both ratios surpass predicted values at the first $\delta^{13}C_{org}$ peak of OAE2, decrease slightly during the Plenus Cold Event and show a large increase during the plateau phase. The remaining interval is characterized by increased values above background level as in the lower part of Core SN°4 prior to the onset of OAE2.

Phosphorus accumulation rates (AR) decline during the onset, peak and plateau phase of OAE2 (Fig. 8) with $P_{total}$ AR, $P_{react}$ AR and CaP AR declining by ~75% and AlFeP AR by ~40 %. Phosphorus accumulation recovers after the end of the plateau phase and increases for $P_{total}$ by ~230 %, for $P_{react}$ and CaP by ~250 % and for AlFeP by ~210 %.

**4 Discussion**

**4.1 Influence of paleogeographic setting and weathering regime**

Changes in the weathering regime of the source area during OAE1a, as shown by the XRF-scanner derived Log(K/Al), influenced the sedimentary record of Cores LB1 and LB3 (Fig. 3). Increased Log(K/Al) and higher $\delta^{18}O$ suggest

predominantly physical weathering and/or intensified erosion, characteristic for drier conditions, and/or markedly seasonal rainfall prior to OAE1a (C2 and early C3) (Figs. 3 and 4). A decrease in Log(K/Al), synchronous with a shift to carbonate-depleted sediments, indicated by Log(Terr/Ca), coincides with an increase in $\delta^{18}O$ associated with the first transient cold event within the onset phase of OAE1a (C4). This major cooling event in the South Provence Basin was probably of global character (e.g., Jenkyns 2018). The covariance of $\delta^{18}O$ and Log(K/Al), in combination with published palynological and geochemical data (e.g., Masure et al., 1998; Hochuli et al., 1999; Keller et al., 2011; Föllmi, 2012; Cors et al., 2015), suggests that the La Bédoule area was located at the northern edge of the subtropical high pressure desert belt and shifted into the northern hemisphere westerlies with increased rainfall during OAE1a. An equatorward contraction of the sub-tropical high-pressure belt also occurred in East Asia during the mid-Cretaceous warm period (Hasegawa et al., 2012). The paleo-location of La Bédoule (Cores LB1 and LB3) close to 30° N (Masse et al., 2000) would predispose the area for latitudinal changes of the Hadley cell with drier conditions in times of an expanded high-pressure belt and increased rainfall resulting from an equatorward contracted Hadley cell. Strengthening fluctuations between drier and wetter climate conditions are recorded during the onset, main and early plateau phase. After a stabilization during the plateau phase, wet conditions prevailed after OAE1a.

The evolution of the weathering regime across OAE2 differed substantially in the Tarfaya Basin (Fig. 2), probably due to the lower paleolatitude and different paleoceanographic setting in a coastal upwelling area. The period prior to OAE2 was characterized by low sea surface temperatures and high mean Log(K/Al) exhibiting high variability, suggesting orbital forcing of a monsoonal hydrological cycle and weathering regime. A first prominent temperature decrease occurred during the late precursor and early onset phase of OAE2, similar to the global temperature decrease during OAE1a (Fig. 3; Jenkyns 2018). A shift to lower mean Log(K/Al), indicative of stronger chemical weathering and/or reduced monsoonal seasonality in the source area coincided with low sea surface temperatures during the precursor phase and suggests increased upwelling intensity and intensified monsoonal wind forcing. A shift back to higher Log(K/Al) indicating weaker chemical weathering in the source area occurred at the end of the plateau phase of OAE2. The main interval of OAE2 (onset, peak and plateau phase) was characterized by intense chemical weathering and a weaker response of the hydrological cycle to orbital forcing, suggesting that the hinterland of the Tarfaya Basin was more or less permanently under the influence of tropical convective rainfall. The recovery phase and the immediate post-OAE2 period was once more characterized by weaker chemical weathering and a stronger response to orbital forcing, typical for a monsoonal regime at the northern edge of the seasonal swing of the ITCZ. Persistently high temperatures (O'Brien et al., 2017) in the recovery and post-OAE2 phase, as indicated by low $\delta^{18}O$ in SN°4, suggest that the equatorward shift of the monsoonal zone and southward expansion of the high-pressure desert belt occurred during a period of global warming. Changes in the weathering regime during the carbon isotope excursion may, thus, have been linked to major fluctuations in atmospheric $CO_2$ associated with increased carbon burial and enhanced response to local insolation forcing in the late onset and peak phase of OAE2 (Fig. 5).

**4.2 Definitions and durations of OAE1a and OAE2**

The estimated durations of OAE1a and OAE2 depend on the definitions of the events (based either on the stratigraphic extent of organic-rich sediments or $\delta^{13}C$ excursions) and vary substantially from minimum estimates of 45 kyr for OAE1b to >3 Myr for OAE1a (Table 1). Whereas there is a broader consensus to define the duration of OAE2 as the entire interval between the onset of the positive $\delta^{13}C$ excursion and the end of the recovery phase, the definition of OAE1a is commonly restricted to the interval of organic-rich sedimentation (Selli-Level). This interval encompasses the onset and peak of the

$\delta^{13}C$ excursion, but does not include the plateau and recovery phase (e.g., Malinverno et al., 2010). At La Bédoule, these differences result in durations of 1.1 Myr (onset and peak phases only) and of 2.7 Myr (including plateau and recovery phases), which is almost four times the duration of OAE2. The differences in the definition of OAE1a and OAE2 stem from the different stratigraphic relationship between black shale accumulation and $\delta^{13}C$ excursion in most of the "classic" localities in central and northern Italy, Britain and North Africa. At these locations, the highest accumulation rates of organic

carbon occurred within the peak and plateau phase of the carbon isotope excursion during OAE2, whereas during OAE1a the duration of black shale deposition was much shorter than the $\delta^{13}C$ excursion.

**4.2.1 OAE1a**

The restriction of the black shale facies (Selli level) to the onset phase of the global carbon isotope curve in many classical OAE1a localities requires an additional carbon sink to account for the globally high $\delta^{13}C$ values in segments C6 and C7, if

we assume that the same mechanism of organic carbon burial is responsible for global positive $\delta^{13}C$ excursions. However, the sedimentary record of black shales and ocean anoxia is patchy across OAE1a. Typical black shales with high TOC values corresponding to the Selli level of OAE1a are not encountered in the stratigraphic succession of La Bédoule. Typical examples such as the Resolution Guyot record (Jenkyns, 1995) and the classical records from Cismon and Piobbico (Erba, 1992; Erba et al., 1999) are likely not fully representative of global organic matter accumulation across OAE1a. Preliminary

TOC data from more complete sedimentary successions across OAE1a in southern Spain (Cau section and the recently drilled Cau core, Naafs et al, 2016; Ruiz-Ortiz et al., 2016) provide evidence for enhanced organic carbon accumulation during carbon isotope segment C7 (Naafs et al., 2016). These sections exhibit unusually high sedimentation rates and a carbon isotope record, which matches that of La Bédoule (Naafs et al, 2016), underlining the completeness of both records. In the northern South Atlantic (e.g., DSDP Site 364 offshore Angola), an expanded sediment sequence showing evidence of

cyclic episodes of intense anoxia/euxinia was deposited at neritic depths in a large, restricted basin (Behrooz et al., 2018; Kochhann et al., 2014), which may have contributed to organic carbon burial during C6 and C7. There is also evidence of black shale facies extending beyond isotope segment C6 in northeastern Tunisia (Elkhazri et al., 2013), albeit with TOC values in the upper Bedoulian black limestone facies (corresponding to segment C7) significantly lower (<0.5 %) than in the black limestones deposited within the C4 segment (4.5 %).

Another possible candidate for enhanced organic carbon burial during the plateau stage of the $\delta^{13}C$ excursion are shallow marine to brackish water sediments in the high Arctic (i.e., Axel Heiberg Island, Canadian Artic Archipelago), where organic carbon rich sedimentation (TOC >5 %) persisted during most of OAE1a (Herrle et al., 2015). However, the most likely location for burial of larger amounts of organic carbon during OAE1a is the central and east Pacific Ocean. Incompletely cored sequences on the Shatsky Rise indicate deposition of organic matter rich sediments during OAE1a under dysoxic-

anoxic conditions at ODP Sites 1207 and 1213 that possibly extended into the upper part of the $\delta^{13}C$ excursion (Dumitrescu and Brassell, 2005, 2006; Dumitescu et al., 2006). Moreover, large areas of lower Cretaceous crust and sediments along the margins of the Pacific Ocean, which are likely candidates for organic matter-rich sedimentation, have been subducted, leaving less than 15 % of the lower Aptian ocean seafloor accessible today (Hay, 2007). Thus, we consider the record of global organic carbon burial documented by the positive $\delta^{13}C$ excursion to be more representative for the duration of OAE1a

than the stratigraphic extent of local black shales. However, this view strongly depends on the interpretation of globally elevated $\delta^{13}C$ values as the result of enhanced organic carbon burial. We cannot fully exclude an influence of shallow marine and terrestrial carbonate cycles in maintaining elevated $\delta^{13}C$ values in the marine dissolved inorganic carbon reservoir (Weissert et al., 1998). For example, an increase in the proportion of carbonate weathering, relative to organic carbon and silicate weathering, could have maintained long lasting positive excursions in marine $\delta^{13}C$ without substantially enhanced

burial of organic carbon (Kump and Arthur, 1999).

    Our estimated durations of OAE1a isotope stages agree with those of Malinverno et al. (2010) for the C6 to C8 stages (Table 2), but deviate from the reconstruction of Scott (2016). Minor differences with the estimates of Malinverno et al. (2010) are caused by differing definitions of boundaries between isotope stages in the Cismon core and the LB3/LB1 composite record and by different calculations of orbital periods. The plateau phase (C7) lasted for 1340 kyr in the LB3/LB1 record, in

agreement with estimates of 1590 kyr by Malinverno et al. (2010) and of 990 kyr by Scott (2016). The duration of 315 kyr for stage C6 agrees with the 349 kyr estimate by Malinverno et al. (2010), but substantially differs from the 110 kyr proposed by Scott (2016). There are larger deviations from the estimates of Malinverno et al. (2010) and Scott (2016) for stages C5 and C4 in the LB3/LB1 record. Isotope stage C5 has a shorter duration of 280 kyr compared to 510 kyr, estimated by Malinverno et al. (2010) but agrees with the 210 kyr duration of Scott (2016). By contrast, the main increase of the

positive carbon isotope excursion, corresponding to C4 (Menegatti et al., 1998), has a duration of 390 kyr in LB3/LB1, which is ~60 % longer than the estimate of 239 kyr by Malinverno et al. (2010) and ~140 % longer than the 160 kyr duration of Scott (2016). The duration of C4 in LB3/LB1 is interpreted as a response to long eccentricity (405 kyr). Strong eccentricity control during the precursor phase (C3) and during the onset phase (C4) is supported by EHA-analysis and power spectra (Figs. 4 , Supplementary Material S3.1 and S3.2), suggesting that orbital eccentricity influenced the global

carbon cycle and regional sedimentation. The largest differences are within the precursor phase (C3) with 440 kyr in LB3/LB1 and 46.7 kyr in the Cismon core reconstructed by Malinverno et al. (2010) and the 80 kyr estimate of Scott (2016), which is probably caused by hiatuses in the Cismon APTICORE record. Malinverno et al. (2010) and Scott (2016) did not

rule out the occurrence of smaller sedimentary gaps in the OAE1a sequence of the Cismon core, which are not long enough to compromise the complete biostratigraphic succession reconstructed by Erba et al. (1999).

**4.2.2 OAE2**

The negative carbon isotope excursion at the onset of OAE2 is absent from many classic OAE2 sections in Europe and US Western Interior Basin due to the stratigraphic incompleteness of these records. The missing negative excursion is commonly associated with a short-term hiatus at the onset of the positive excursion, often expressed as a sharp lithological contact (base of the Bonarelli horizon in some of the Umbrian Scaglia sections (Italy) (Jenkyns et al., 2007; Batenburg et al.,

2016) and sub-plenus erosion surface in the Eastbourne Section (UK) (Paul et al., 1999; Gale et al., 2005)). However, negative spikes preceding the onset of the positive $\delta^{13}C$ excursion were documented in high resolution data sets, even in the relatively condensed Umbrian sections (e.g., Furlo section, Jenkyns et al., 2007) and in more expanded shelf sections at Wunstorf, northern Germany (Voigt et al., 2008) and Oued Mellegue, Tunisia (Nederbragt and Fiorentino, 1999). Records from expanded sections in Mexico (Elrick et al., 2009) and Japan (Nemoto and Hasegawa, 2011) also clearly exhibit the

negative excursion. Recently, a high resolution $\delta^{13}C$ record from the South Pacific Ocean and cyclostratigraphic age model based on magnetic susceptibility measurements indicated that the duration of the negative excursion was ~50 kyr (Li et al., 2017b; Gangl et al., 2019) and allowed correlation of the onset of the negative isotope shift with the beginning of LIP activity at ~94.44 ±0.14 Ma (Du Vivier et al., 2015). These estimates (Table 3) for the precursor phase agree with our new durations of 75 kyr for the last two prominent $\delta^{13}C_{org}$ minima in Core SN°4. The onset phase lasted for 68 kyr in core SN°4,

broadly in agreement with 30 ±13 kyr estimated by Gangl et al. (2019) for the Gongzha section (Tibet) and 110 ±25 kyr for stage 3a in the Sawpit Gully section (New Zealand) of Li et al. (2017b). There is a larger discrepancy for the peak phase with 86 kyr reconstructed for the Tarfaya Basin and 170 ±25 kyr or 200 ±25 kyr for the sections in Tibet or New Zealand, respectively, possibly caused by undetected short-term sedimentation rate changes connected to climatic changes during the Plenus Cold Event. Estimates for the duration of the plateau and recovery phase vary widely between the three localities

possibly arising from difficulties defining the inflection point at the beginning and end of the plateau phase. The plateau phase lasted for 204 kyr in the Tarfaya Basin, 370 ±25 kyr in the Tibet and 660 ±25 kyr in the Sawpit Gully section. The newly reconstructed duration of the recovery phase in the Tarfaya Basin is 435 kyr, contrasting the much shorter durations of the Gongzha (170 ±25 kyr) and Sawpit Gully (40 ±25 kyr) sections. The agreement in the cumulative length of plateau and recovery phase underlines the difficulties in the defining and reconstructing durations for the latest part of OAE2.

**4.2.3 Amplitude of $\delta^{13}C$ excursions**

The amplitude of carbon isotope excursions is generally higher for organic matter than for bulk carbonate (Jenkyns, 2010). This is the case for OAE2 in the Tarfaya Basin, where the amplitude is ~4 ‰ for organic $\delta^{13}C$ and ~2.5 ‰ for carbonate $\delta^{13}C$ across the basin (Kuhnt et al., 1986, 1990, 2005, 2017; Kolonic et al., 2005; Tsikos et al., 2004). In the Tarfaya Basin,

the high organic content is of marine origin and changes in $\delta^{13}C$ values, thus, mainly reflect global reservoir changes at intermediate values of local productivity (Van Bentum et al., 2012). However, marked regional differences in the amplitude of the organic and bulk inorganic $\delta^{13}C$ excursions were also reported for OAE2 (Van Bentum et al., 2012; Wendler 2013; Kuhnt et al., 2017). Possible causes include local differences in the $\delta^{13}C$ of dissolved inorganic carbon dependent on local productivity, oxygenation and diagenesis (Kuypers et al., 1999; Jenkyns, 2010; Van Bentum et al., 2012; Kuhnt et al., 2017). Similar differences in the amplitude of $\delta^{13}C$ excursions are recorded for OAE1a: highest amplitudes reach or slightly exceed 4 ‰ in both organic carbon and carbonate from the western Tethys margin (e.g., Menagatti et al., 1998), whereas at deep water localities with distinct black shale deposition of the Selli event the amplitude remains lower (~2.5 ‰, Malinverno et al., 2010).

A long-term increase in $\delta^{13}C$ over the Cretaceous and early Cenozoic was reported by Katz et al. (2005) and attributed to the evolution of large-celled phytoplankton and increased organic carbon burial efficiency in expanding depositional spaces (e.g., shallow shelves around the Atlantic), created during the rifting phase of the current Wilson cycle (Wilson, 1966; Worsley et al., 1986). Continuous removal of isotopically light organic carbon by deposition in shallow seas depleted the global carbon reservoir. The OAE1a and OAE2 records show differences in the background levels between the preceding intervals and after the terminations of both events with 0.8 ‰ in $\delta^{13}C_{carb}$ for OAE1a and 0.3 ‰ in $\delta^{13}C_{org}$ for OAE2. An overall increase over OAE2 is also reconstructed for deposits of the English chalk (Jarvis et al., 2006). We assume that the long-term increase observed by Katz et al. (2005) was stepwise and to a large degree influenced by Cretaceous OAEs. The large-scale isotopic depletion of the global carbon reservoir accompanying these major disturbances of the global carbon cycle was not entirely compensated during the recovery phase and in the multimillion-year interval following these events inducing a long-term increase in $\delta^{13}C$ during the Cretaceous.

### 4.3 Impact of orbital forcing on the evolution of OAE1a and OAE2

The orbital configuration favouring enhanced marine biological productivity in low latitudes may have been different for OAE1a and OAE2. Recent studies (e.g., Batenburg et al., 2016; Kuhnt et al., 2017) tuned the onset of OAE2 to a 405 kyr eccentricity maximum that succeeded an extended period of low seasonality, caused by a 2.4 Myr eccentricity minimum, which would be associated with stronger obliquity forcing during the precursor phase. The much shorter onset phase of OAE2 suggests a faster and stronger response of the ocean-climate system to the carbon cycle perturbation, either related to a shorter and more intense initial carbon dioxide release or to different orbital configuration such as higher amplitude eccentricity cycles. Additionally, higher rates of sea level rise would have promoted enhanced organic carbon burial in shallow shelf seas. These extensive dysoxic to anoxic shelf seas would have favoured fast recycling of limiting nutrients, thereby enhancing primary production and further boosting organic carbon burial. Eccentricity control on both sea-level and organic carbon burial is also suggested by the ~90 and ~600 kyr durations of the peak phases of OAE2 and OAE1a (maximum organic carbon deposition and highest $\delta^{13}C$ values) (Figs. 4 and 5).

By contrast, a strong obliquity imprint is detected during the plateau phases (Figs. 4 and 5). A period of high amplitude obliquity forcing would have steepened the latitudinal temperature gradient and intensified atmospheric circulation (Batenburg et al., 2016; Kuhnt et al., 2017), thus helping to maintain elevated biological productivity in the tropical oceans. Obliquity-forced intensification of monsoonal systems may have resulted in periods of enhanced tropical weathering associated with enhanced nutrient supply to the ocean and wind driven equatorial upwelling, which promoted carbon sequestration during the plateau and recovery phases. However, recycling of essential nutrients from buried sediments or release from new sources (e.g., flood basalts of LIPs) would have been necessary to maintain this new equilibrium state over extended periods of time. The plateau stage lasted seven times longer during OAE1a than during OAE2, which excludes orbital forcing as a primary control on its duration. This is also supported by differences in the response to orbital forcing at the end of the plateau phases: precession and obliquity imprint during OAE1a contrast with a persistent obliquity signal during OAE2.

In addition to precessional variability, changes in orbital obliquity were a critical forcing factor of climate oscillations during the recovery phase of both OAEs. Obliquity determines the summer intertropical insolation gradient, which was recently suggested as an important driver of changes in tropical and subtropical hydrology and sedimentation patterns (Bosmans et al., 2015). An increased insolation gradient between the tropical summer and winter hemisphere during high obliquity leads to intensified atmospheric circulation within the Hadley cell, resulting in stronger cross-equatorial winds and intensified moisture transfer into the summer hemisphere. Continental climate proxy data and models suggest a contraction of the Hadley cell and a latitudinal shift of the subtropical high-pressure belt towards the equator during mid-Cretaceous super-greenhouse conditions (Hasegawa et al., 2012; Hay and Flögel, 2012). This climate scenario would have placed the hinterland of the Tarfaya Basin in the dry, hot subtropical desert climate zone during OAE2, whereas the La Bédoule area was likely influenced by the humid zone of the Northern Hemisphere westerlies during OAE1a. An increase of cross-equatorial winds and intensified rainfall at obliquity maxima would, thus, have affected the Tarfaya Basin by increasing upper ocean mixing and upwelling and the La Bédoule area by increased rainfall due to intensified westerlies.

Periods of high amplitude 41 kyr obliquity variations last ~400-800 kyr and are separated by nodes of weak obliquity forcing with a duration of ~200-400 kyr (Supplementary Material S5.1, Laskar et al., 2004). During the recovery phase of OAEs, obliquity-forced low latitude climate oscillations may have led to periods of enhanced equatorial upwelling, which would have promoted carbon sequestration and, over several hundred thousand years, depleted the ocean's nutrient pool. A period of low variability in orbital obliquity (obliquity node), commonly associated with global cooling episodes in Cenozoic climate records (e.g., Pälike et al., 2006), may have ultimately terminated OAE1a and 2.

## 4.4 Role of phosphorus recycling in maintaining high productivity during OAEs

Phosphorus is the primary limiting nutrient controlling marine biological productivity on longer (geological) timescales (e.g., Holland, 1978; Broecker and Peng, 1982; Smith, 1984; Codispoti, 1989) with the potential to control the occurrence of high productivity events (e.g., Föllmi 1996; Handoh and Lenton, 2003). By contrast to nitrate, which can be synthesized from

atmospheric nitrogen primarily by cyanobacterial $N_2$ fixation under anoxic conditions (e.g., Rigby and Batts, 1986; Rau et al., 1987; Kuypers et al., 2004), the phosphorus supply to the ocean is restricted by riverine terrestrial input (Ruttenberg, 2003). This constitutes a limiting factor for increased marine primary productivity. Thus, alternative nutrient sources such as enhanced terrestrial input or recycling of marine sediments were necessary to sustain high primary productivity over the extended durations of Cretaceous OAEs (e.g., Nedebragt et al., 2004).

$C_{org}/P_{react}$ and $N_{total}/P_{react}$ close to or above the Redfield ratio were measured within the MCE and OAE2 intervals, suggesting preservation of the initial atomic ratio of primary production, possibly even enhanced by a change in the phytoplankton community (Geider and La Roche, 2002) or depletion in $P_{react}$. Unusually high phosphorus values in the intervals preceding the MCE and during the onset of OAE2 were likely caused by starved sedimentation, which may be due to formation of hard grounds and condensed sections at sea-level high stands or winnowing of fine-grained sediment by intensified bottom currents.

Regional changes in redox conditions towards anoxic or euxinic conditions in the lower water column would have inhibited fixation of remobilized phosphorus in authigenic CaP, thus allowing leakage of phosphorus into the water column and resulting in increased sedimentary C:P and N:P. Intensification of oxygen depletion during the plateau phase of OAE2 is associated with increasing $C_{org}/P_{react}$ and $N_{total}/P_{react}$ due to reduced diagenetic phosphorus precipitation permitting phosphorus leakage from the sediment into the water column. However, phosphorus accumulation rates started to decline earlier during the precursor phase of OAE2. A trend to more reducing conditions during the plateau and recovery phase of OAE2 was also suggested by Kolonic et al. (2005), based on high accumulation rates of redox-sensitive elements. A similar sequence of events was reconstructed by Stein et al. (2011) for OAE1a in deposits from the Gorgo a Cerbara section (Umbria-Marche basin) in central Italy (Supplementary Material S11). Phosphorus accumulation and $C_{org}/P_{total}$ increased during the precursor phase (C3 of Menegatti et al., 1998) and decreased during the onset phase (C4 of Menegatti et al., 1998), suggesting enhanced phosphorus recycling from the sediments, as for OAE2 in Core SN°4 (Fig. 8 and Supplementary Material S11). Stein et al. (2011) also reported a minor increase of $C_{org}/P_{total}$ at the base of the peak phase (C5+C6) and persistently high $C_{org}/P_{total}$ during the late peak phase (C5+C6), as for OAE2 in Core SN°4.

An earlier study by Poulton et al. (2015), focusing on Fe speciation proxies during the onset, peak and early plateau phase of OAE2 in nearby drill core S57, found cyclic variations between euxinic and ferruginous conditions. The stratigraphically extended interval (from the MCE to early Turonian) investigated at lower resolution by Scholz et al. (2019) is characterized by a high proportion of unpyritized reactive Fe in the total Fe pool. The results of this study are consistent with a proxy signature that is indicative of anoxic and non-sulfidic, so-called ferruginous, water column conditions throughout the studied interval (Poulton and Canfield, 2011). However, Scholz et al. (2019) argued that dissolved Fe (and hydrogen sulphide) concentrations in the water column of the Tarfaya system were unlikely higher than those observed in modern upwelling zones (e.g., Peru margin) on account the low terrigenous sedimentation rates and tropical weathering on the adjacent continent. Previous studies proposed that phosphorus burial might be enhanced under ferruginous conditions, implying a negative feedback for the oceanic phosphorus pool and primary production (e.g., März et al., 2008). However, Scholz et al.

(2019) did not observe a close relationship between Fe and P burial, despite a ferruginous signature in the sediments, which supports the notion that dissolved Fe concentrations and rates of Fe oxide precipitation in the Tarfaya Basin were moderate and overall similar to modern upwelling systems (Wallmann et al., 2019).

The impact of oxygen depletion on the release of phosphorus into the water column was shown by recent studies on modern OMZs (e.g., Noffke et al., 2012; Schollar-Lomnitz et al., 2019) with implications for increased primary productivity through feedback mechanisms. Phosphorus recycling from the sediments may have sustained high primary productivity over extended periods of time, thus, contributing to the long duration of OAE1a and OAE2. Phosphorus remobilization under anoxic conditions from vast areas of flooded shelf sediments facilitated high organic carbon burial rates during the onset, peak and plateau phases of the three anoxic events, and may have acted as a positive feedback process, enhancing carbon burial and removal of light carbon isotopes from the marine dissolved inorganic carbon reservoir and resulting in positive carbon isotope excursions.

## 5 Conclusions

New high-resolution stable isotope and XRF-scanner data were integrated with published records from Cores LB1 and LB3 in the South Provence Basin (Lorenzen et al., 2013; Moullade et al., 2015) and from Core SN°4 in the Tarfaya Basin (Kuhnt et al., 2017; Beil et al., 2018) to contrast the temporal evolution of two of the most significant Oceanic Anoxic Events: OAE1a and OAE2. The structure of the marine $\delta^{13}C$ records suggests a similar evolution of the carbon cycle during both OAEs, although the duration of the individual phases differed substantially. Both OAEs exhibit negative excursions (precursor phase) with a duration of ~430 kyr (OAE1a) and ~130 kyr (OAE2), immediately preceding the onset of the positive carbon isotope excursion. The onset phases, lasting ~390 kyr for OAE1a and ~70 kyr for OAE2, was characterized by intervals of rapid $\delta^{13}C$ increase separated by small plateaus with slower $\delta^{13}C$ change. Prominent cooling events recorded as $\delta^{18}O$ increases started during the latest onset phase and extended into the peak phase (Plenus Cold Event during OAE2), which lasted ~600 kyr during OAE1a and only ~90 kyr during OAE2. The plateau phase extended over ~1340 kyr for OAE1a but lasted only ~200 kyr for OAE2. However, the durations of the recovery phases were similar with ~380 kyr (OAE1a) and ~440 kyr (OAE2). The different durations of the precursor and plateau phases of OAE1a and OAE2 may have been linked to the magnitude and duration of the triggering volcanic exhalations. In addition, different orbital configurations may have influenced long-term marine organic carbon burial on a global scale.

Phosphorus speciation data from Core SN°4 in the Tarfaya Basin provide new insights into the dynamics of this essential nutrient during the MCE and OAE2. Phosphorus speciation shows a predominance of Ca-bound phosphorus surpassing concentrations of Al/Fe-bound P by one to two orders of magnitude. Phosphorus bound to Al- and Fe-oxyhydroxides is elevated during the early Cenomanian (305-273 m) in Core SN°4, reflecting sedimentation in a shallower environment or less intensive redox-induced, early diagenetic cycling of iron oxyhydroxides. Elevated ratios $C_{org}/P_{react}$ and $N_{total}/P_{react}$ during the MCE and OAE2 indicate a change in the water column towards more reducing conditions. Oxygen-free bottom water

permitted the leakage of dissolved phosphorus from the sedimentary column and increased the ratios of $C_{org}$ and $N_{total}$ to $P_{react}$ within the sediments. This change is apparently synchronous with enhanced organic carbon burial in the Tarfaya Basin. The delayed increase of $C_{org}/P_{react}$ and $N_{total}/P_{react}$ with respect to the $\delta^{13}C_{org}$ increase suggests that the Cenomanian OAEs were not initiated by shelfal phosphorus remobilization. However, the coincidence of maximum organic carbon burial and highest $C_{org}/P_{react}$ and $N_{total}/P_{react}$ underlines the significance of phosphorus leakage from sediments for maintaining high organic carbon burial rates necessary to sustain globally recognized $\delta^{13}C$ shifts during OAEs over extended time periods.

## Data availability

Newly acquired data from this study are available at the Pangaea Data Repository.

## Author contribution

Sebastian Beil, Wolfgang Kuhnt and Ann Holbourn designed the study and wrote the manuscript. Wolfgang Kuhnt, Ann Holbourn and Mohamed Aquit planned and supervised the drilling of Cores LB1, LB3 and SN°4. Data for this study were acquired by Sebastian Beil, Mohamed Aquit, Janne Lorenzen, Julian Oxmann, Florian Scholz, Klaus Wallmann, Ann Holbourn and Wolfgang Kuhnt. Time series analysis was performed by Sebastian Beil. El Hassane Chellai facilitated fieldwork and drilling. All authors read and provided comments on the manuscript.

## Competing interests

The authors declare that they have no conflict of interest.

## Acknowledgments

We would like to thank Nils Andersen (Leibniz Laboratory for Radiometric Dating and Stable Isotope Research, Kiel) and Michael Joachimski (GeoZentrum Nordbayern, Friedrich-Alexander University of Erlangen-Nürnberg, Erlangen) for stable isotope measurements, Dieter Garbe Schönberg and Samuel Müller (Institute of Geosciences (Institute of Geosciences, Christian Albrechts-University, Kiel) for X-ray fluorescence scanning advice and Anke Bleyer, Bettina Domeyer and Regina Surberg (Geomar, Kiel) for laboratory assistance. This project was supported by the German Research Foundation (DFG) through Subproject A7 of the Collaborative Research Center (SFB) 754 (Climate-Biogeochemistry Interactions in the Tropical Oceans) and the Emmy Noether Research Group ICONOX (Iron Cycling in Continental Margin Sediments and the Nutrient and Oxygen Balance of the Ocean). We would like to gratefully acknowledge Hugh Jenkyns, Christian März and Matthew Clarkson for insightful reviews that significantly improved the manuscript.

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

**Table 1.** Durations of Cretaceous OAEs.

| Carbon isotope excursion | Time interval | Duration | Estimate for | Publication |
|---|---|---|---|---|
| OAE1a | early Aptian to late Aptian | ~1.0–1.3 Myr | carbon isotope stages C3 to C6 of Menegatti et al. (1998) in $\delta^{13}C$ | Li et al. (2008) |
| | | 1.11 ±0.11 Myr | Selli Level | Malinverno et al. (2010) |
| | | 1.36 Myr | increase to inflection point of $\delta^{13}C$ | Scott (2014) |
| | | 1.16 Myr | carbon isotope stages C3 to C6 of Menegatti et al. (1998) | Moullade et al. (2015) |
| OAE1b | late Aptian to early Albian | 210 kyr | positive isotope excursion in $\delta^{13}C$ | Erbacher et al. (2001) |
| | | 46 kyr | black shale deposition | |
| | | 45 kyr | negative excursion and increase to $\delta^{13}C$ maximum | Wagner et al. (2007) |
| | | 40 kyr | positive isotope excursion in $\delta^{13}C$ | |
| | | 1.84 Myr | | Scott (2014) |
| OAE1c | middle Albian | 1.01 Myr | increase to $\delta^{13}C$ maximum | Scott (2014) |
| OAE1d | Albian – Cenomanian boundary interval | ~400 kyr | increase to $\delta^{13}C$ maximum | Petrizzo et al. (2008) |
| | | ~1 Myr | ACBI (Albian-Cenomanian boundary interval) from the onset of peak A (OAE1d) to the return to background values after peak D in $\delta^{13}C$ | Bornemann et al. (2017) |
| | | 1.21 ±0.17 Myr | OAE1d from the onset of peak A to peak C in $\delta^{13}C$ | Gambacorta et al. (2019) |
| | | 233 kyr | onset and peak phase in $\delta^{13}C$ | Yao et al. (2018) |
| | | ~400 kyr | increase to inflection point in $\delta^{13}C$ | Scott (2014) |
| MCE | middle Cenomanian | ~200 kyr | onset to second peak in $\delta^{13}C$ | Voigt et al. (2004) |
| | | <400 kyr | onset to second peak in $\delta^{13}C$ | Reboulet et al. (2013) |
| | | 210 kyr | onset until return to background values in $\delta^{13}C$ | Eldrett et al. (2015) |
| OAE2 | Cenomanian – | 563–601 kyr | onset, peak and plateau phase in $\delta^{13}C$ | Sageman et al. (2006) |

| | Turonian boundary interval | 847–885 kyr | onset, peak, plateau phase and recovery phase in $\delta^{13}C$ | |
|---|---|---|---|---|
| | | 450–500 kyr | in S13 (Tarfaya Basin) between onset and end of plateau in $\delta^{13}C$ | Meyers et al. (2012a) |
| | | 500–550 kyr | in 1261B (Demerara Rise) between onset and end of plateau in $\delta^{13}C$ | |
| | | 516–613 kyr | in Angus Core (Western Interior Seaway); onset, peak, plateau phase and recovery phase in $\delta^{13}C$ | Ma et al. (2014) |
| | | 559–675 kyr | in Portland Core (Western Interior Seaway); onset, peak, plateau phase and recovery phase in $\delta^{13}C$ | |
| | | 520 kyr | onset phase in $\delta^{13}C$ | Scott (2014) |
| | | 710 ±170 kyr | onset until end of plateau phase in $\delta^{13}C$ | Eldrett et al. (2015) |
| | | 920 ±170 kyr | negative excursion, onset, peak and plateau phase in $\delta^{13}C$ | |
| | | ~900 kyr | onset until end of plateau phase in $\delta^{13}C$ | Jenkyns et al. (2017) |
| | | 650 ±25 kyr | onset until end of plateau phase in $\delta^{13}C$ | Li et al. (2017b) |
| | | 820 kyr | onset, peak, plateau phase and recovery phase in $\delta^{13}C$ | |
| | | 675 kyr | onset until end of plateau phase in $\delta^{13}C$ | Charbonnier et al. (2018) |
| | | 956 kyr | onset, peak, plateau phase and recovery phase in $\delta^{13}C$ | |
| | | 930 ±25 kyr | onset until end of plateau phase in $\delta^{13}C$ | Gangl et al. (2019) |

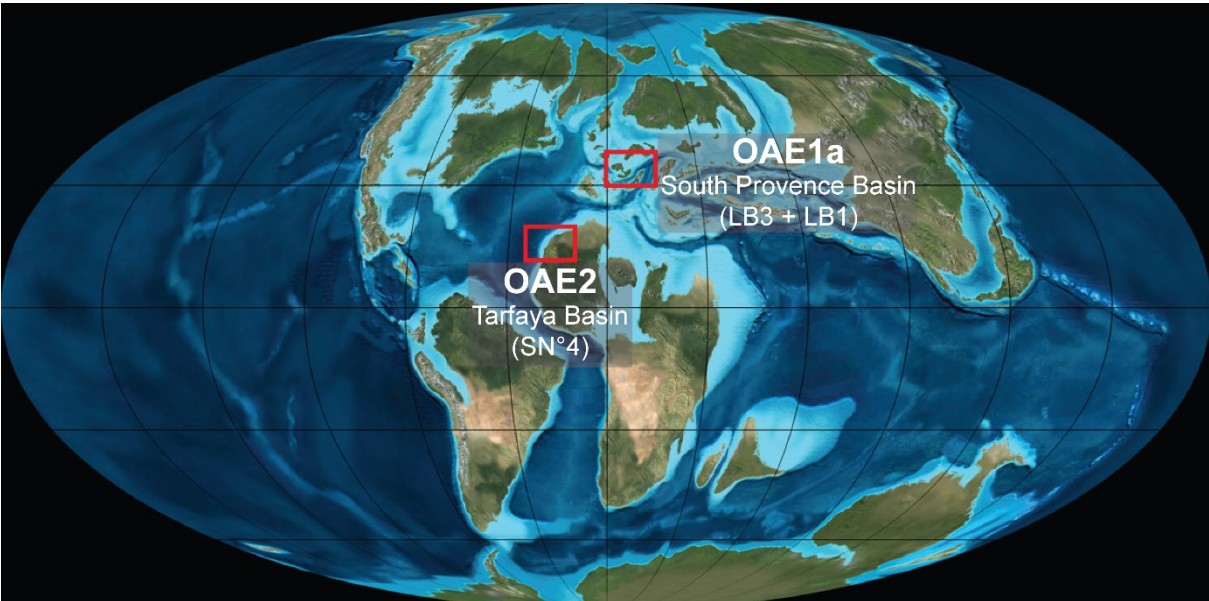

5 **Fig. 1.** Reconstruction of Late Cretaceous paleogeography at 100 Ma (Ron Blakey, Colorado Plateau Geosystems). Red boxes indicate locations of South Provence and Tarfaya Basins.

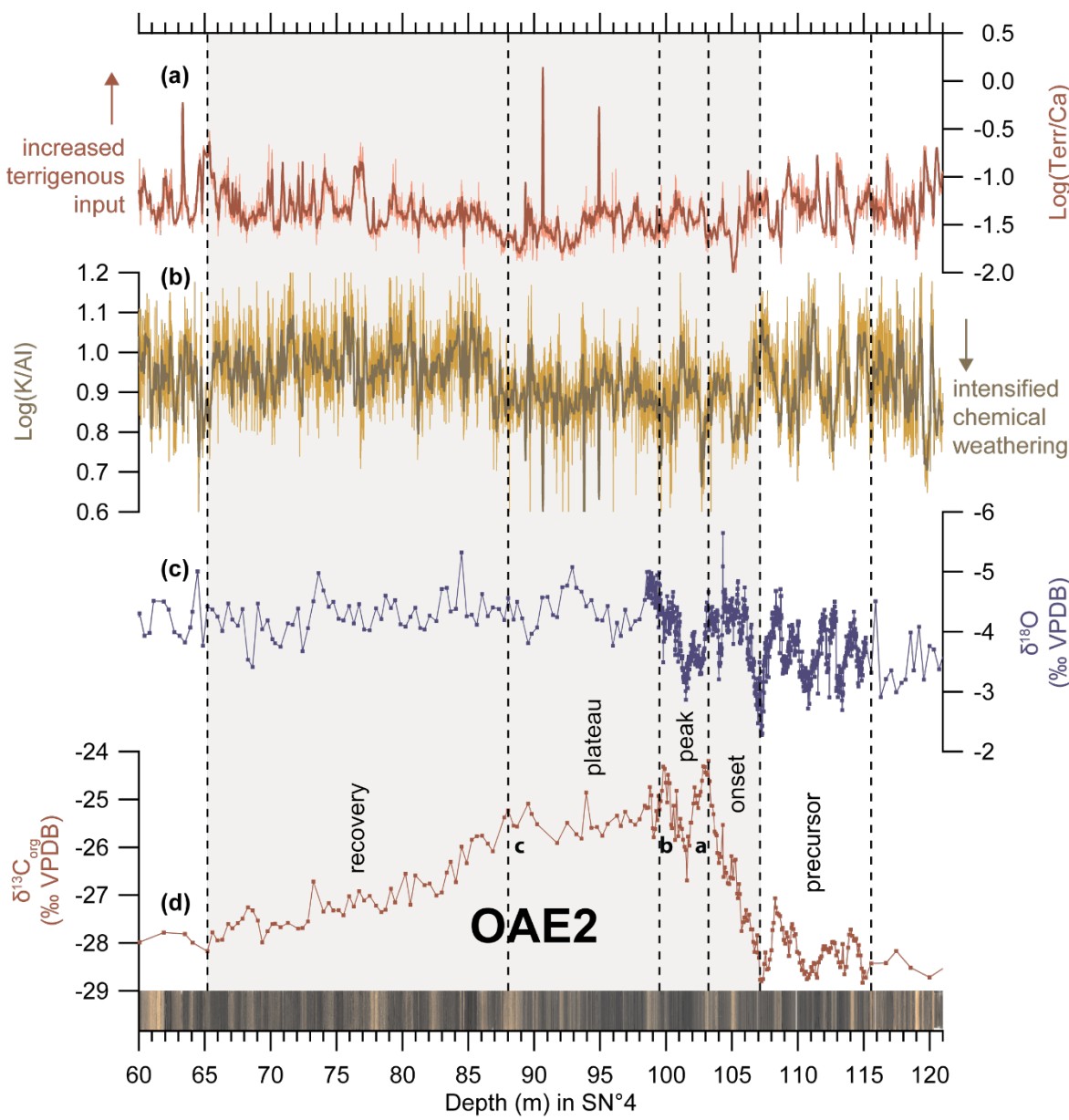

**Fig. 2.** Line scan photographs, stable isotope and XRF-scanning elemental records spanning OAE2 in Tarfaya Basin Core SN°4. **(a)** Log(Terr/Ca) from Kuhnt et al. (2017) and Beil et al. (2018). **(b)** Log(K/Al) from Kuhnt et al. (2017) and Beil et al. (2018). **(c)** $\delta^{13}O$ and **(d)** $\delta^{13}C_{org}$ from Kuhnt et al. (2017) and Beil et al. (2018); a-c indicate prominent maxima following Voigt et al. (2007).

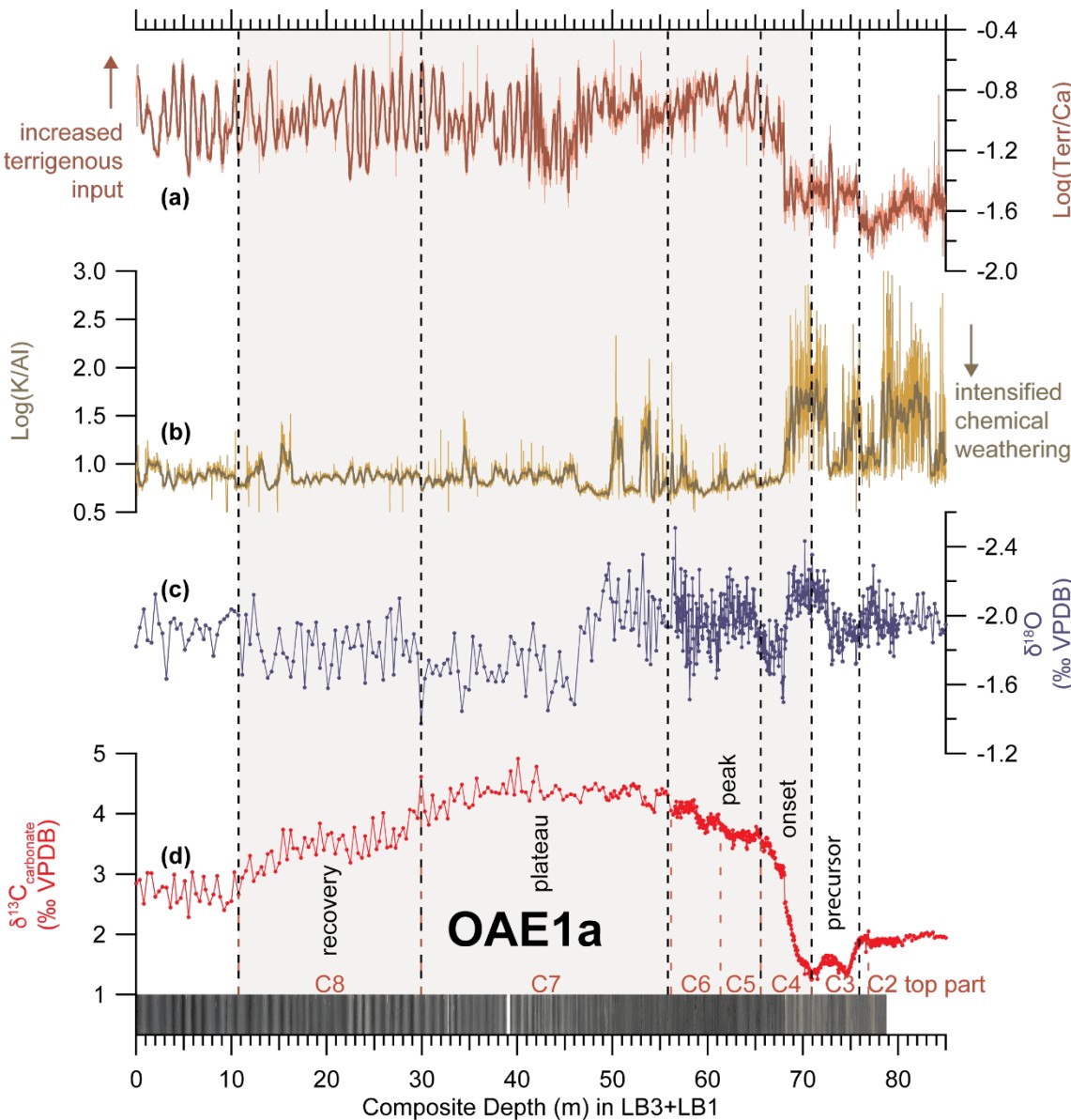

**Fig. 3.** Line scan photographs, stable isotope and XRF-scanning elemental records spanning OAE1a from La Bédoule Cores LB3/LB1. **(a)** Log(Terr/Ca). **(b)** Log(K/Al). **(c)** $\delta^{13}O$ and **(d)** $\delta^{13}C_{carbonate}$ from Lorenzen et al. (2013) and Moullade et al. (2015) with new high-resolution isotope data from this study. The boundaries between individual phases (precursor, onset, peak, plateau, recovery) of OAE1a are indicated by dashed black lines. Carbon isotope stages C2-C8 of Menegatti et al. (1998) are indicated by red lines.

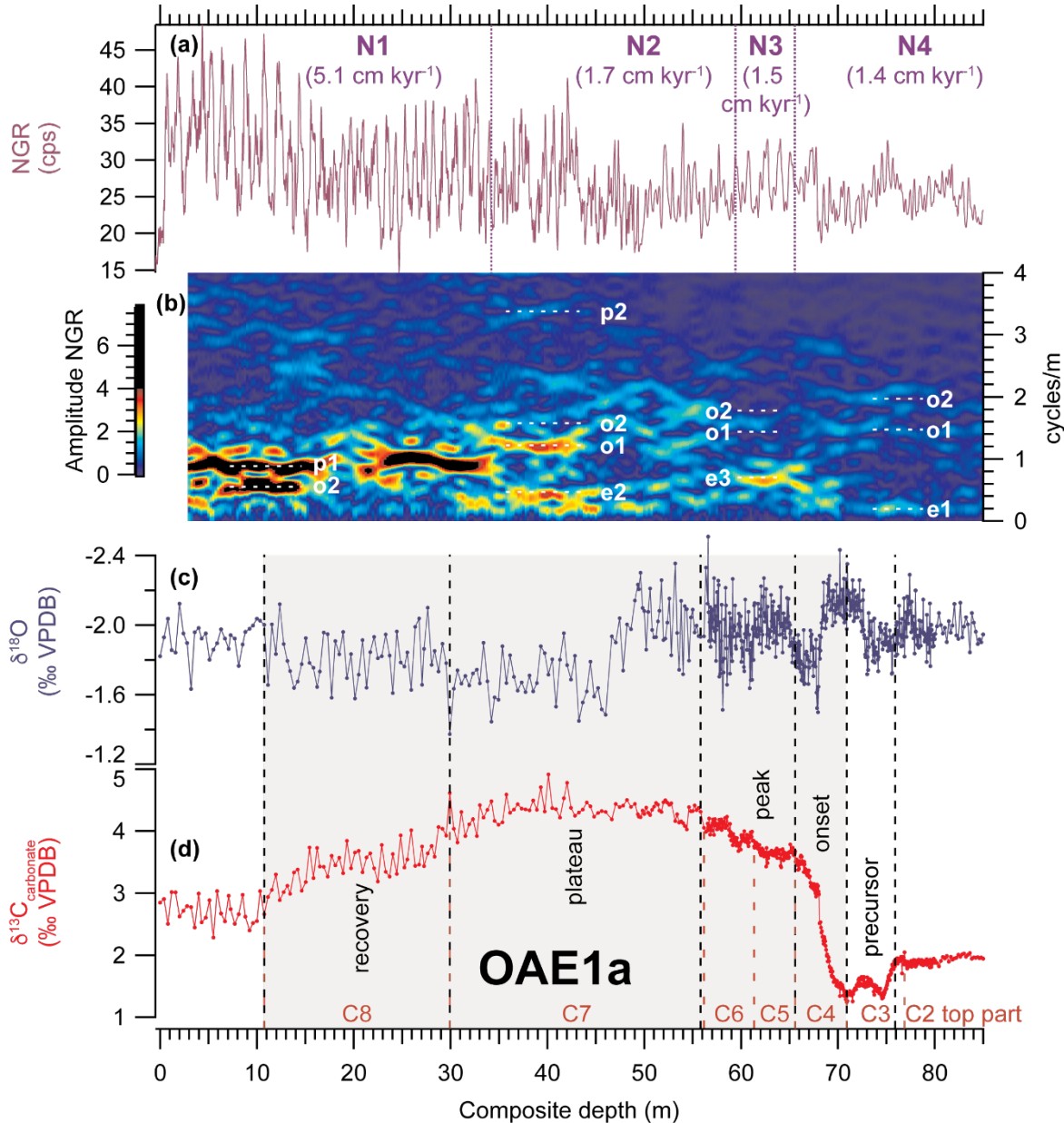

**Fig. 4.** Identification of orbital periodicities across OAE1a using Evolutive Harmonic Analysis (EHA) of Natural Gamma Ray (NGR) data in La Bédoule Cores LB3/LB1. **(a)** NGR; purple dashed lines mark boundaries between segments N1-N4 (Supplementary Material S3 and S3.2); mean sedimentation rates given in brackets. **(b)** EHA plot of NGR; orbital periodicities are indicated by white dashed lines; eccentricity e1 (404 kyr), e2 (124 kyr) and e3 (95 kyr), obliquity o1 (47 kyr) and o2 (37 kyr), precession p1 (22 kyr) and p2 (18 kyr) (Supplementary Material S5.2). **(c)** $\delta^{18}O$ and **(d)** $\delta^{13}C_{carbonate}$ from Lorenzen et al. (2013) and Moullade et al. (2015) with new high-resolution isotope data from this study. Carbon isotope stages C2-C8 of Menegatti et al. (1998) are indicated by red dashed lines. Individual phases (precursor, onset, peak, plateau, recovery) of OAE1a are separated by black dashed lines. Note: the EHA color scheme is limited to 50% of the maximum amplitude to enhance visibility in the lower part of the core.

**Table 2.** Durations and sedimentation rates for Aptian C-stages (following Menegatti et al., 1998) in La Bédoule Core LB3/LB1 and comparison with durations from Malinverno et al. (2010) and Scott (2016).

| C-stage | Top in LB3/LB1 (m) | Base in LB3/LB1 (m) | Segment | Duration (kyr) | Mean SedRate during C-stage (cm kyr$^{-1}$) | Durations (kyr) following Malinverno et al. (2010) | Durations (kyr) of Scott (2016) |
|---|---|---|---|---|---|---|---|
| C8 | 10.75 | 29.92 | N1 | 377 | 5.09 | | |
| C7 | 29.92 | 56.16 | N1 + N2 | 1343 | 1.95 | 1590 | 990 |
| C6 | 56.16 | 61.35 | N2 + N3 | 315 | 1.65 | 349 | 110 |
| C5 | 61.35 | 65.58 | N3 | 281 | 1.50 | 510 | 210 |
| C4 | 65.58 | 70.91 | N4 | 388 | 1.37 | 239 | 160 |
| C3 | 70.91 | 76.87 | N4 | 434 | 1.37 | 46.7 | 80 |

**Table 3.** Durations of individual phases for OAE2 in Tarfaya Basin Core SN°4 and comparison with Li et al. (2017b) and Gangl et al. (2019).

| Phases | SN°4 (Tarfaya Basin, SW Morocco; this study) | Gongzha section (Tibet; Li et al., 2017b) | Sawpit Gully section (New Zealand; Gangl et al., 2019) |
|---|---|---|---|
| Precursor | 126 kyr (last two prominent minima: 75 kyr) | 50 ±25 kyr (stage 2) | 50 kyr |
| Onset | 68 kyr | 110 ±25 kyr (stage 3a) | 30 ±13 kyr |
| Peak | 86 kyr | 170 ±25 kyr (stages 3b+c) | 200 ±25 kyr |
| Plateau | 204 kyr | 370 ±25 kyr (stage 4) | 660 ±25 kyr |
| Recovery | 435 kyr | 170 ±25 kyr (stage 5) | 40 ±25 kyr |

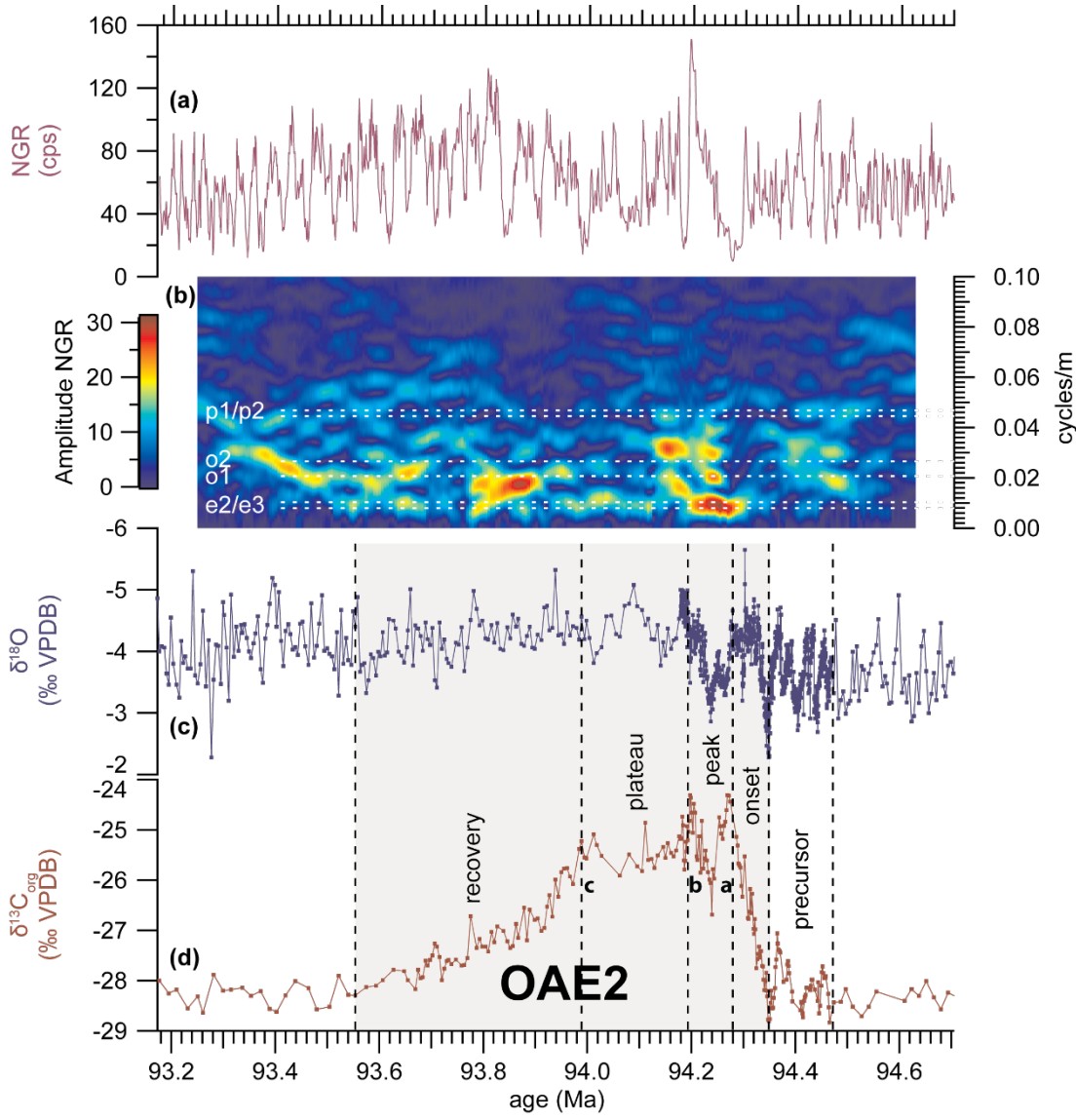

**Fig. 5.** Identification of orbital periodicities over OAE2 from Evolutive Harmonic Analysis (EHA) of Natural Gamma Ray (NGR) data in Tarfaya Basin Core SN°4. **(a)** NGR. **(b)** EHA plot of NGR; orbital periodicities are indicated by white dashed lines: eccentricity e2 (127 kyr) and e3 (97 kyr), obliquity o1 (48 kyr) and o2 (38 kyr), precession p1 (22 kyr) and p2 (18 kyr) (Supplementary Material 5.3). **(c)** $\delta^{18}O$ and **(d)** $\delta^{13}C_{org}$ from Kuhnt et al. (2017) and Beil et al. (2018); a-c indicate prominent maxima following Voigt et al. (2007).

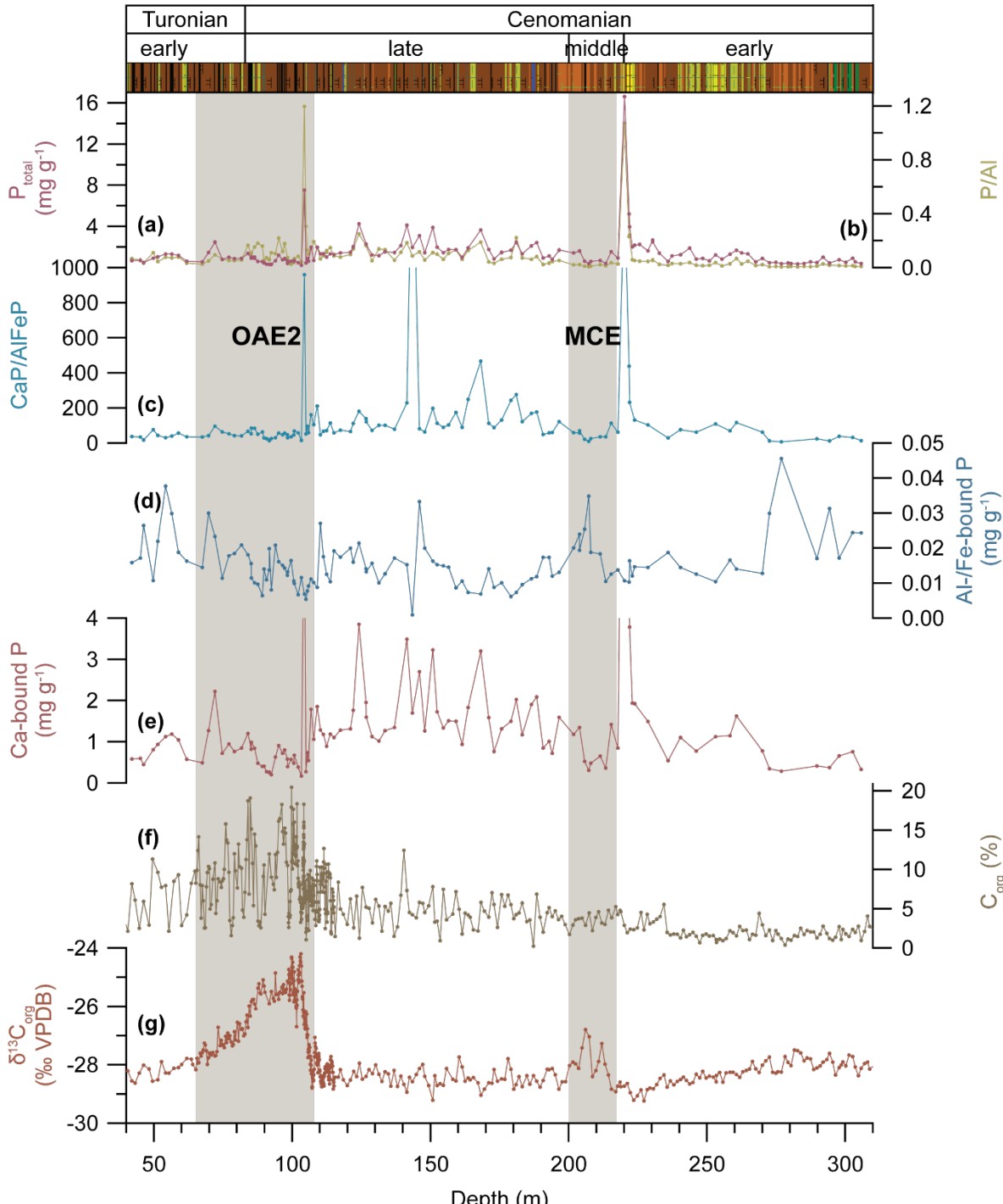

50    **Fig. 6.** Phosphorus concentration and speciation in Tarfaya Basin Core SN°4. **(a)** Concentration of total phosphorus ($P_{total}$). **(b)** ratio of total phosphorus to aluminium (P/Al) measured on bulk sediment. **(c)** ratio of calcium- to aluminium/iron-bound phosphorus (CaP/AlFeP). **(d)** concentration of aluminium/iron bound-phosphorus (Al-/Fe-bound P). **(e)** concentration of calcium-bound phosphorus (Ca-bound P). **(f)** concentration of organic carbon ($C_{org}$) and **(g)** $\delta^{13}C_{org}$ from Kuhnt et al. (2017) and Beil et al. (2018).

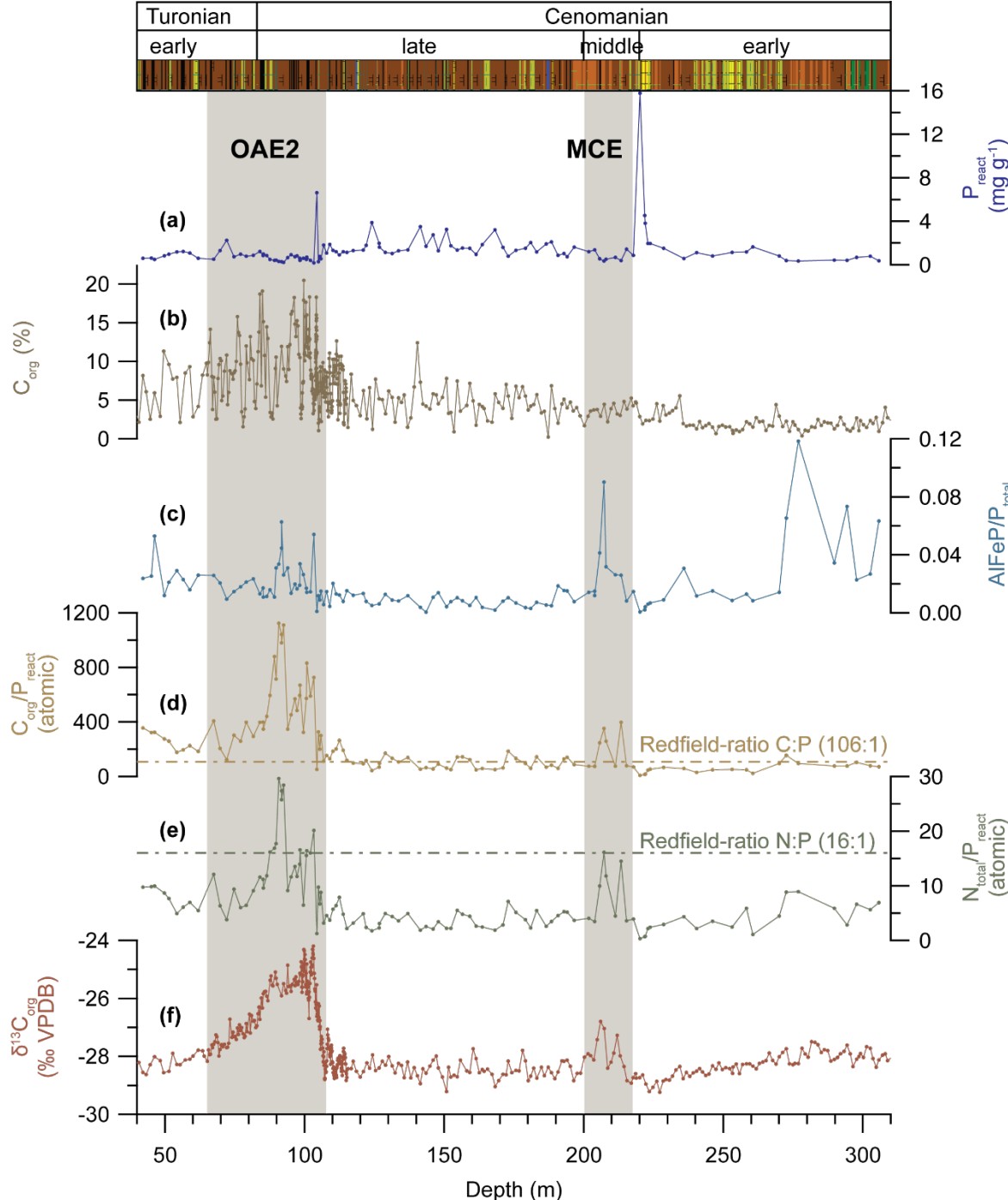

**Fig. 7.** Phosphorus speciation in Tarfaya Basin Core SN°4. **(a)** concentration of reactive phosphorus calculated as the sum of Al-/Fe- and Ca-bound P ($P_{react}$). **(b)** concentration of organic carbon ($C_{org}$). **(c)** ratio of aluminium and iron bound phosphorus to total phosphorus (AlFeP/$P_{total}$). **(d)** atomic ratio of organic carbon to reactive phosphorus ($C_{org}/P_{react}$) with Redfield ratio 106:1 indicated as dashed line. **(e)** atomic ratio of total nitrogen to reactive phosphorus ($N_{total}/P_{react}$) with Redfield ratio 16:1 indicated as dashed line. **(f)** $\delta^{13}C_{org}$ from Kuhnt et al. (2017) and Beil et al. (2018).

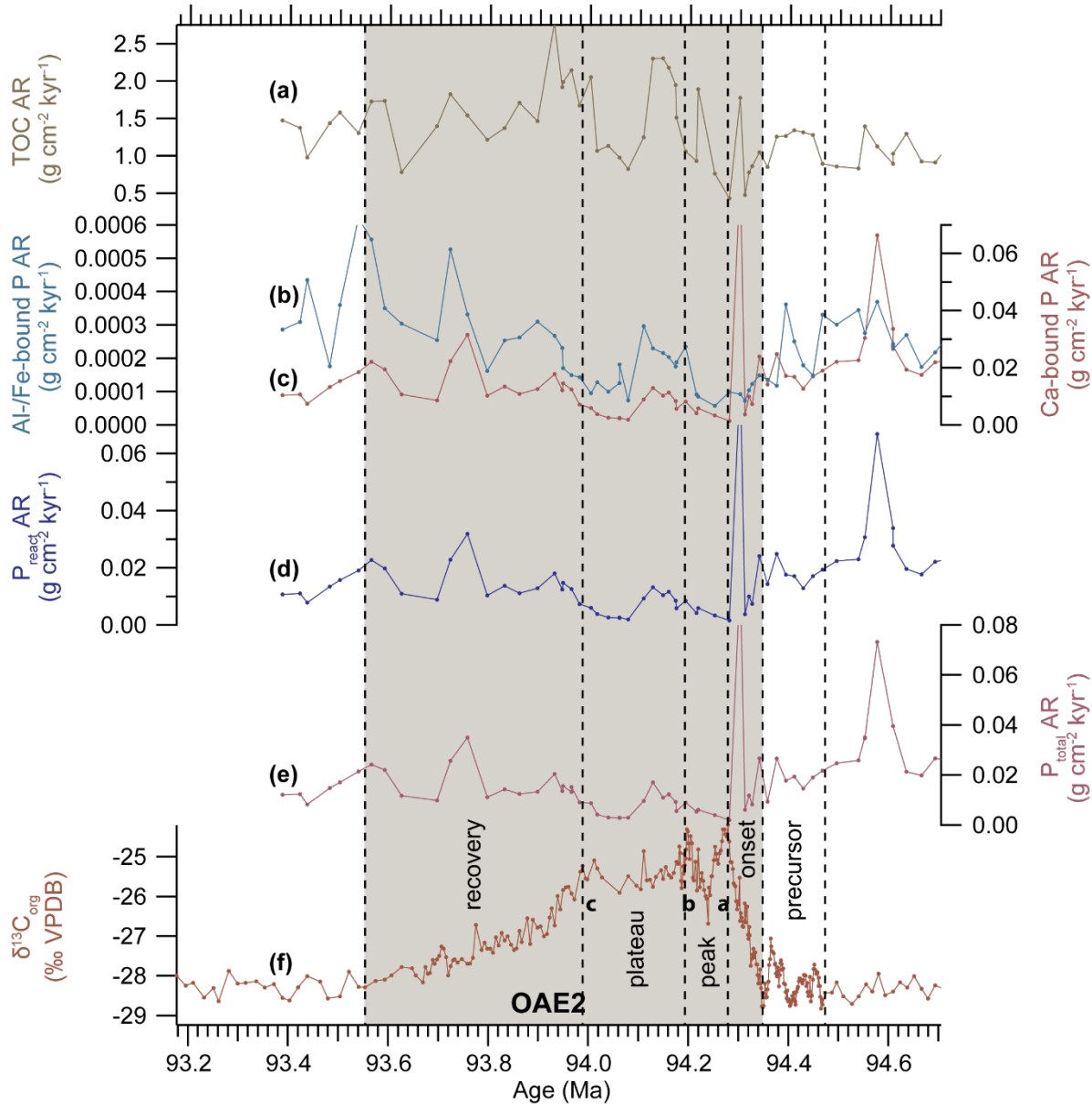

**Fig. 8.** Phosphorus accumulation rates across OAE2 in Tarfaya Basin Core SN°4 (age model based on Meyers et al. 2012a). **(a)** Accumulation rate of organic carbon (TOC AR). **(b)** accumulation rate of aluminium and iron bound phosphorus (Al-/Fe-bound P AR). **(c)** accumulation rate of calcium bound phosphorus (Ca-bound P AR). **(d)** accumulation rate of reactive phosphorus calculated as the sum of Al-/Fe- and Ca-bound P ($P_{react}$ AR). **(e)** accumulation rate of total phosphorus ($P_{total}$ AR). **(f)** $\delta^{13}C_{org}$. Stable Isotope data and organic carbon concentrations from Kuhnt et al. (2017) and Beil et al. (2018); a-c indicate prominent maxima following Voigt et al. (2007).

60

65

