# Peer review of "Cretaceous Oceanic Anoxic Events prolonged by phosphorus cycle feedbacks"

_Climate of the Past, 2019_

## Referee Comment (RC1) · Hugh Jenkyns (Referee) · 17 Oct 2019

This paper aims to illustrate the potential role of P-cycle feedbacks in prolonging OAE 1a and OAE 2. The paper contains useful data and some excellent diagrams but is rather densely written, skips over some important problems, switches tenses a lot when describing geological phenomena, and ignores some relevant literature. The fundamental point that the low P/TOC ratios in the OAE sediments points definitively to phosphorus recycling (and nutrient re-supply to planktonic biota) during these events tends is easily lost. The issue of P recycling during OAEs has, of course, been made previously, including from a modelling perspective (e.g. Mort papers; Nederbragt et al). The value of the account lies in the fact that the sections described are stratigraphically very expanded and give superb detail as to changes in the carbon cycle before, during

and after OAEs.

Abstract and beyond: the statement that the evolution of the carbon-isotope curve of the two OAEs, as classically defined, shows remarkable similarities needs to be qualified. The defining characteristic of OAE 2 is the overarching positive excursion; for OAE 1a it's the negative excursion. Many OAE 2 sequences (e.g. Eastbourne, UK) show no negative excursion, although its absence is probably due to the presence of the sub-plenus erosion surface in the case of the English section. More needs to be made of all this because the apparently more stratigraphically complete Tarfaya record of OAE 2 clearly offers a unique perspective. The New Zealand record of Gangl et al. (EPSL, 518, 172–182) and Japanese record of Nemoto and Hasegawa (Palaeo-cubed, 309, 271–280) may also show this negative excursion but it is certainly not everywhere apparent. As regards OAE 1a, as illustrated in Fig. 4, the main positive excursion extends higher than the C6 segment (1.e. post OAE 1a - unless C6 is extended higher in the section). Do we need a total redefinition of OAE 1a, as implied here? If so, all of this needs to be made clear as perhaps we have been biased by the records of the Cismon and Piobbico cores. But there is a problem: where are the abundant black shales that correspond to the C6 and C7 relatively heavy carbon-isotope segments, given that the original OAE definition is rooted in the quasi-coeval organic-rich record on a global basis?

Line 21: not clear which events are being referred to with 'respectively'

Lines 23–25: nutrients may have been supplied by basalt–seawater interaction, probably involving LIPs. (Mentioned later in the text but not here)

Line 55: cite original paper by Scholle and Arthur (1980)

Line 66: Are these Mort papers the appropriate references for discussion of transgression? See Jenkyns (1980)

Line 92: rewrite as: 'A variety of phosphorus species are discriminated against in these

sediments.

Line 98: change 'In contrast' to 'By contrast'

Line 131: hyphenate 'intermediate-resolution' to read as written here

Line 164: do you mean nannofossils and planktonic foraminifera? 'Shells' rather implies macrofossils.

Line 189: hyphenate 'metal-free' to read as written here

Line 271: change 'In contrast' to 'By contrast'

Line 280: state in which segments of the OAE 1a record the cooling events have been identified. Do they conform to those illustrated in Jenkyns, 2018 (Phil .Trans Roy. Soc.) from multiple localities, namely: C3, C4 and C6? Which cooling events in the OAE 2 record correspond with the Plenus Cold Event? Are these multiple events registered anywhere else? Do they relate to the fact that Tarfaya was a palaeo-upwelling site with upward movements of cooler water or are they global? The largest positive oxygen-isotope shift (Fig. 2) seems to predate the rise in carbon isotopes: i.e. before major global carbon burial was registered, which is not as stated in the text (line 284).

Line 318: it would be worth looking at the C-segment durations given by Scott , 2016: (Barremian–Aptian–Albian carbon isotope segments as chronostratigraphic signals: numerical age calibration and durations. Stratigraphy, 13, 21–47) to see how they compare with your data.

Line 329: hyphen not necessary in 'orbitally tuned'

Lines 343–345 and Fig. 2 and Fig. 5: it might be useful to label the features on the OAE 2 carbon-isotope profile (a,b,c,d), as illustrated by Voigt et al., 2017, EPSL. 53, 196–210.

Line 465: 'prevail' - this is present tense and is but one example where past tense should be used for geological narrative. There are many instances of this error in the

text. It's also important to maintain clarity when moving from description of an isotope curve to inferences about the environment.

Line 500: compare with the durations given by Scott (see above)

Line 504: change 'In contrast' to 'By contrast'

Line 552: change 'In contrast' to 'By contrast'

Lines 579: Mention needs to be made of the key paper by Handoh and Lenton, 2003 (Global biogeochemical Cycles, 17, 1092, who also discuss the cycling of phosphorus to maintain productivity during OAEs. This paper draws on the important papers of Föllmi (Geology, 1995, 23, 503-506; Earth-Science Reviews, 1996, 40, 55–124) that discuss the long-term stratigraphy of phosphorus in the stratigraphic record.

Line 581: say how synthesized from atmospheric nitrogen. This will involve a brief discussion on cyanobacteria and papers by Kuypers et al. (Geology, 2004), and others

Page 615: is 'largest' the right word? Most significant?

Line 622: given that the durations of the carbon-isotope plateau phases are so different, is their causality different as well? We know that the plateau phase of OAE 2 corresponds with maximum organic-carbon burial, at least in the Tethys–Atlantic region - but there is no such evidence for OAE 1a (except possibly Shatsky Rise). So what is going on?

HC Jenkyns

---

## Short Comment (SC1) · 24 Oct 2019

Dear Babette, dear Sebastian and co-authors, let me first say that I don't tend to write reviews that I haven't been invited to. I do not mean to make the authors' lives harder than they already are. However, the topic your nice manuscript is about is quite close to my heart, and I have therefore decided to add a few comments that might help to widen the perspective of the manuscript and put into context of a few publications that the authors might have missed. As it happens, some of these publications are (co-) authored by me and my review could be understood as shameless self-promotion. This is not my intention, but the editor might have a different view on this and may therefore decide to ignore my comment.

The manuscript prepared by Beil et al. is an impressive data set on an impressive number samples from two locations that resolve two OAEs (1a and 2) in very high temporal resolution. I have read the comment by Hugh Jenkyns, which focuses on the definition/duration/isotopic expression of the OAEs, and I will not go into any detail on those. Instead, my comment refers to the phosphorus side of the story.

I applaude the authors for having generated a very nice P speciation data set, for reporting the recovery of their extractions relative to total P, and for a very detailed method description in the appendix. In the broadest sense, I also agree with the interpretation of the authors that P recycling from the seafloor during much of OAE2 has potentially led to higher primary productivity, fueling an anoxia-productivity feedback loop that has been previously suggested to extend the "lifetime" of OAEs.

My comments, which are all included in the attached PDF as annotations, relate to the (a) a more precise distinction between different redox conditions (namely ferruginous versus euxinic) and (b) the weathering regime. The main reason for raising these issues is that Poulton et al. (2015) conducted a study on the onset of OAE2 from a different Tarfaya core, with a focus on the potential effects of weathering conditions on land on ocean redox, and the related response of the P cycle to these redox changes. Since this manuscript is using very similar methods and proxies on samples from effectively the same location, I think it would be an omission and a missed opportunity to not refer to the published manuscript, and put the new data into context. My comments in the PDF are hopefully self-explanatory, but please feel free to ask for clarificaton.

I hope the authors will take my relatively minor comments in the good spirit of scientific exchange, and I am looking forward to seeing the final version of the manuscript published in Climate of the Past.

Kind regards, Christian Maerz

Please also note the supplement to this comment:

https://www.clim-past-discuss.net/cp-2019-118/cp-2019-118-SC1-supplement.pdf

[Figure]

**Supplement:**

[revised manuscript text omitted]

---

## Referee Comment (RC2) · Matthew Clarkson (Referee) · 13 Nov 2019

The manuscript presents an impressive dataset of P-speciation data and high resolution XRF core scans, which help build on earlier works regarding i) the duration of OAEs, and ii) the hypothesis for P-cycling as an important feedback mechanism for OAE development. The work involved in this manuscript could feasibly represent two papers, if the authors saw fit, as the cyclo-stratigraphy aspect over-shadows the P speciation and the data presentation become very lengthy. I have read through the detailed comments from Hugh Jenkyns and Cristian Maerz and agree with their inputs. I will try to give additional contributions, rather than repeating their observations. Generally, this is an impressive dataset and it shouldn't take much work to address these comments.

General Comments:

I think there is a missed opportunity here in that one of these cores has been extensively studied previously by the authors (Scholz et al., 2019), with Fe-speciation, redox sensitive metals (Mo, V) and N isotopes. The new P-speciation data would complement this previous study very nicely and more could be made of integrating the two datasets. This could be valuable for the discussion of redox and P cycling through OAE2 and would help give more contextual information, particularly with reference to the evolving nature of redox conditions through the core. I think it would be very useful to the community to examine P-speciation results within the context of the established redox framework that varies locally from nitrogenous to euxinic, and compare this to intervals of ferruginous and euxinic deposition elsewhere (Poulton et al., 2015), and so I am somewhat mirroring a comment made by Dr. Maerz.

Minor comments:

Line 93: 'oceanic anoxia'

Line 114: The MCE is referred to frequently as it appears in the records, however not much background is given on the significance of this event. Please detail if this is a local feature or a global event comparable to the other OAEs studied.

Line 160: As a disclaimer, I am not so familiar with XRF core scanning techniques, but I would be suspicious of using Fe as a terrestrial element, included in the logTerr/Ca proxy, as there is likely redox-dependent behaviour in these settings that would obscure or bias trends in terrestrial elements if Fe is included. It might be that Fe is lost from the sediment due to reduction in the pore-waters (thereby removing any Fe cycles), or that Fe has been enriched through Fe-shuttling across the basin. It would be possible that the stepped increase in Terr/Ca could be caused by an increase in Fe, due to enrichment of highly reactive Fe-phases (e.g. at the onset of OAE1a). It is also possible that this could create apparent cyclicity, analogous to the cyclicity in FeHR/FeT in other Tarfaya data (Poutlon et al, 2015). If Fe is plotted separately or removed from this

measure, do you see any behaviour that might be indicative of local redox changes dominating the record?

This could be an opportunity to add additional information on redox systematics...Can you pull out Fe/Al from the XRF data to aid comparison to the Fe-speciation cyclicity observed by Poulton et al., 2015 in the other Tarfaya core and the previous Fe-speciation data of Scholz et al., 2019?

Also, what about the dilution effect of Ca from high organic carbon production, would this potentially create cycles or stepped changes through the OAEs. There seems to cycles in TOC from just looking at the linescan photograph, so how much of the cyclicity in logTerr/Ca can be explained by simply changing CaCO3 concentration?

Could you also please clarify what NGR represents in terms of sedimentary components that drives the cyclicity, and how this links to the orbital pacing mechanisms.

Line 196: is smoked the correct term for this? ashed?

Line 280: the PCE is often associated with faunal changes that represent different water mass movements or local re-oxygenation. I think it is a bit misleading to focus on the extinction aspect. More could be done to reference other studies here.

---

## Author Comment (AC1) · 1 Jan 2020

Reply to Interactive comment on "Cretaceous Oceanic Anoxic Events prolonged by phosphorus cycle feedbacks" by Sebastian Beil et al. Hugh Jenkyns (Referee) RC1 hughj@earth.ox.ac.uk

This paper aims to illustrate the potential role of P-cycle feedbacks in prolonging OAE 1a and OAE 2. The paper contains useful data and some excellent diagrams but is rather densely written, skips over some important problems, switches tenses a lot when describing geological phenomena, and ignores some relevant literature. The fundamental point that the low P/TOC ratios in the OAE sediments points definitively to

phosphorus recycling (and nutrient re-supply to planktonic biota) during these events tends is easily lost. The issue of P recycling during OAEs has, of course, been made previously, including from a modelling perspective (e.g. Mort papers; Nederbragt et al). The value of the account lies in the fact that the sections described are stratigraphically very expanded and give superb detail as to changes in the carbon cycle before, during and after OAEs.

First of all, we would like to sincerely thank Hugh Jenkyns for his insightful and constructive feedback. Following his comments, we revised and streamlined the manuscript to improve readability and highlight the main findings of our study. We also addressed important questions, which were not previously touched upon, and we included missing essential references (listed at the end of this rebuttal).

Abstract and beyond: the statement that the evolution of the carbon-isotope curve of the two OAEs, as classically defined, shows remarkable similarities needs to be qualified.

We will discuss in more detail the similarities and dissimilarities in the evolution of the carbon-isotope curves of the two OAEs in the revised manuscript. We plan to modify the abstract and to insert the following text into section 4.2.2 of the discussion:
The similarities in the general shape of the $\delta$13C excursion of OAE1a and OAE2 (precursor, onset, peak and plateau phase) suggest similar forcing and response mechanisms. However, there are also remarkable differences in the amplitude and duration of individual phases: in particular, the higher amplitude and extended duration of the precursor phase (negative $\delta$13C excursion preceding the onset of the positive $\delta$13C excursion) and the exceptionally long duration of the plateau phase of OAE1a, which in most classic localities is not associated with the deposition of organic carbon rich black shales ("Selli level"). The different durations of the precursor and plateau phases of OAE1a and OAE2 may have been linked to the magnitude and duration of the triggering volcanic exhalations. In addition, different orbital configurations may have influenced long-term marine organic carbon burial on a global scale. Furthermore, obliquity-forced

intensification of monsoonal systems may have resulted in periods of enhanced tropical weathering associated with nutrient supply to the ocean and wind driven equatorial up-welling, which promoted carbon sequestration during the plateau and recovery phases. Periods of low 41 kyr variability in orbital obliquity (obliquity nodes), which occur every 1.2 Myr and are commonly associated with global cooling episodes in Cenozoic warm climate records (e.g., Pälike et al., 2006), may have triggered interruptions or termination of globally enhanced carbon burial.

The defining characteristic of OAE 2 is the overarching positive excursion; for OAE 1a it's the negative excursion. Many OAE 2 sequences (e.g. Eastbourne, UK) show no negative excursion, although its absence is probably due to the presence of the sub-plenus erosion surface in the case of the English section. More needs to be made of all this because the apparently more stratigraphically complete Tarfaya record of OAE 2 clearly offers a unique perspective. The New Zealand record of Gangl et al. (EPSL, 518, 172–182) and Japanese record of Nemoto and Hasegawa (Palaeo-cubed, 309, 271–280) may also show this negative excursion but it is certainly not everywhere apparent.

To address this comment, we propose to insert a new paragraph in section 4.2.1 of the discussion as follows:
The negative carbon isotope excursion at the onset of OAE2 is absent from many classic OAE2 sections in Europe and US Western Interior Basin due to the stratigraphic incompleteness of these records. The missing negative excursion is commonly associated with a short-term hiatus at the onset of the positive excursion, often expressed as a sharp lithological contact (base of the Bonarelli horizon in some of the Umbrian Scaglia sections (Italy) (Jenkyns et al., 2007; Batenburg et al., 2016) and sub-plenus erosion surface in the Eastbourne Section (UK) (Paul et al., 1999; Gale et al., 2005)). However, negative spikes preceding the onset of the positive $\delta$13C excursion were documented in high resolution data sets, even in the relatively condensed Umbrian sections (e.g., Furlo section, Jenkyns et al., 2007) and in more

expanded shelf sections at Wunstorf, northern Germany (Voigt et al., 2008) and Oued Mellegue, Tunisia (Nederbragt and Fiorentino, 1999). Records from expanded sections in Mexico (Elrick et al., 2009) and Japan (Nemoto and Hasegawa, 2011) also clearly exhibit the negative excursion. Recently, a high resolution $\delta$13C record from the South Pacific Ocean and cyclostratigraphic age model based on magnetic susceptibility measurements indicated that the duration of the negative excursion was âĹij50 kyr (Gangl et al., 2019) and allowed correlation of the onset of the negative isotope shift with the beginning of LIP activity at 94.44 $\pm$0.14 Ma (Du Vivier et al., 2015).

As regards OAE 1a, as illustrated in Fig. 4, the main positive excursion extends higher than the C6 segment (1.e. post OAE 1a - unless C6 is extended higher in the section). Do we need a total redefinition of OAE 1a, as implied here? If so, all of this needs to be made clear as perhaps we have been biased by the records of the Cismon and Piobbico cores. But there is a problem: where are the abundant black shales that correspond to the C6 and C7 relatively heavy carbon-isotope segments, given that the original OAE definition is rooted in the quasi-coeval organic-rich record on a global basis?

We agree that this issue needs to be further discussed. We plan to expand the discussion on the discrepancy between the stratigraphic extension of OAE1a black shale occurrences and the positive carbon isotope excursion (section 4.2.1), as follows:
The restriction of the black shale facies (Selli level) to the onset phase of the global carbon isotope curve in many classical OAE1a localities requires an additional carbon sink to account for the globally high $\delta$13C values in segments C6 and C7, if we assume the same mechanism of organic carbon burial is responsible for global positive $\delta$13C excursions. Typical black shales with high TOC values corresponding to the Selli level of OAE1a are not encountered in the stratigraphic succession at La Bédoule.

However, the sedimentary record of black shales and ocean anoxia is patchy across OAE1a; typical examples such as the Resolution Guyot record (Jenkyns, 1995) and the classical records from Cismon and Piobbico (Erba, 1992; Erba et al., 1999) are probably not fully representative of global organic matter accumulation across OAE1a. Preliminary TOC data from more complete sedimentary successions across OAE1a in southern Spain (Cau section and the recently drilled Cau core, Naafs et al, 2016; Ruiz-Ortiz et al., 2016) provide evidence for enhanced organic carbon accumulation during carbon isotope segment C7 (Naafs et al., 2016, Supplementary Figure S4). These sections exhibit unusually high sedimentation rates and a carbon isotope record, which matches that of La Bédoule (Naafs et al., 2016, Supplementary Figure S1), underlining the completeness of both records. In the northern South Atlantic (e.g., DSDP Site 364 offshore Angola), an expanded sediment sequence showing evidence of cyclic episodes of intense anoxia/euxinia was deposited at neritic depths in a large, restricted basin (Behrooz et al., 2018; Kochhann et al., 2014), which may have contributed to organic carbon burial during C6 and C7. There is also evidence of black shale facies extending beyond isotope segment C6 in northeastern Tunisia (Elkhazri et al., 2013), albeit with TOC values in the upper Bedoulian black limestone facies (corresponding to segment C7) significantly lower (< 0.5 percent) than in the black limestones deposited within the C4 segment (4.5 percent).

Another possible candidate for enhanced organic carbon burial during the plateau stage of the $\delta$13C excursion are shallow marine to brackish water sediments in the high Arctic (i.e., Axel Heiberg Island, Canadian Artic Archipelago), where organic carbon rich sedimentation (TOC >5 percent) persisted during most of OAE1a (Herrle et al., 2015). However, the most likely location for burial of larger amounts of organic carbon during OAE1a is the central and east Pacific Ocean. Incompletely cored sequences on the Shatsky Rise indicate deposition of organic matter rich sediments during OAE1a under dysoxic-anoxic conditions at ODP Sites 1207 and 1213 that possibly extended into the upper part of the $\delta$13C excursion (Dumitrescu and Brassell, 2005, 2006; Dumitrescu et al., 2006). Moreover, large areas of lower Cretaceous crust

and sediments along the margins of the Pacific Ocean, which are likely candidates for organic matter-rich sedimentation, have been subducted, leaving less than 15 percent of the lower Aptian seafloor accessible today (Hay, 2007). Thus, we consider the record of global organic carbon burial documented by the positive $\delta$13C excursion to be more representative for the duration of OAE1a than the stratigraphic extent of local black shales. However, this view strongly depends on the interpretation of globally elevated $\delta$13C values as the result of enhanced organic carbon burial. We cannot fully exclude an influence of shallow marine and terrestrial carbonate cycles in maintaining elevated $\delta$13C values in the marine dissolved inorganic carbon reservoir (Weissert et al., 1998). For example, an increase in the proportion of carbonate weathering, relative to organic carbon and silicate weathering, could have maintained long lasting positive excursions in marine $\delta$13C without substantially enhanced burial of organic carbon (Kump and Arthur, 1999).

Line 21: not clear which events are being referred to with 'respectively'

We deleted this word and revised the sentence. The corrected sentence now reads: Based on analysis of cyclic sediment variations, we estimate the duration of individual phases within OAE1a and OAE 2. We identify: (1) a precursor phase (negative excursion) lasting 430 kyr for OAE1a and 130 kyr for OAE 2, (2) an onset phase of 390 and 70 kyr, (3) a peak phase of 600 and 90 kyr, (4) a plateau phase of 1400 and 200 kyr and (5) a recovery phase of 630 and 440 kyr.

Lines 23–25: nutrients may have been supplied by basalt–seawater interaction, probably involving LIPs. (Mentioned later in the text but not here)

We now mention this possible nutrient source as follows:
The extended durations of the peak, plateau and recovery phases imply fundamental changes in global nutrient cycles either (1) by submarine basalt-sea water interactions, (2) through excess nutrient inputs to the oceans by increasing continental weathering

and river discharge or (3) through nutrient-recycling from the marine sediment reservoir.

Line 55: cite original paper by Scholle and Arthur (1980)

We added this reference:
The ensuing positive $\delta$13C excursion is generally attributed to enhanced burial rates of 12C enriched organic carbon in marine organic-rich shales and/or in terrestrial peat and coal deposits (e.g., Scholle and Arthur, 1980; Jenkyns, 1980; Schlanger et al., 1987; Arthur et al., 1988).

Line 66: Are these Mort papers the appropriate references for discussion of transgression? See Jenkyns (1980)

We included the primary citation of Jenkyns (1980) and we extended the sentence to include the additional source of increased terrestrial weathering, as follows:
These include fertilization by nutrient input in the ocean system in association with the activity of large igneous provinces (LIPs) (e.g., Schlanger et al., 1981; Larson, 1991; Trabucho-Alexandre et al., 2010), sea level controlled remobilization of nutrients from flooded low altitude land areas associated with major marine transgressions (e.g., Jenkyns, 1980; Mort et al., 2008), increased phosphorus input resulting from intensified weathering on land (e.g., Larson and Erba, 1999; Poulton et al., 2015), or release of phosphorus as a main limiting nutrient from sediments into the water column under anoxic bottom water conditions (e.g., Ingall and Jahnke, 1994; Slomp and Van Cappellen, 2007).

Line 92: rewrite as: 'A variety of phosphorus species are discriminated against in these sediments.

We revised the sentence as suggested.

Line 98: change 'In contrast' to 'By contrast'

We revised the sentence as suggested.

Line 131: hyphenate 'intermediate-resolution' to read as written here

Changed

Line 164: do you mean nannofossils and planktonic foraminifera? 'Shells' rather implies macrofossils.

We revised the sentence as follows:
By contrast, calcium (Ca) is assumed to be of marine origin, mainly originating from calcareous nanno- and microplankton.

Line 189: hyphenate 'metal-free' to read as written here

Changed

Line 271: change 'In contrast' to 'By contrast'

Changed

Line 280: state in which segments of the OAE 1a record the cooling events have been identified. Do they conform to those illustrated in Jenkyns, 2018 (Phil .Trans Roy. Soc.) from multiple localities, namely: C3, C4 and C6? Which cooling events in the OAE 2 record correspond with the Plenus Cold Event? Are these multiple events registered anywhere else? Do they relate to the fact that Tarfaya was a palaeo-upwelling site with upward movements of cooler water or are they global? The largest positive oxygen-isotope shift (Fig. 2) seems to predate the rise in carbon isotopes: i.e. before major global carbon burial was registered, which is not as stated in the text (line 284).

The major cooling events that occurred during C4 and C6 correspond to the global events illustrated by Jenkyns (2018). A minor cooling of probable regional character (Jenkyns, 2018) is also evident during stage C3. We propose to expand section 3.2 and to add relevant references as follows:
The $\delta 18O$ curves share common trends, despite cyclic lithological changes in the upper part of the sedimentary record of OAE1a in LB3. Transient cooling events, identified by

$\delta$18O increases in LB3/LB1 and SN°4, occur during the early phases of both OAE1a and OAE2 (Figs. 2, 3 and S11).

A first prominent cooling event prior to the onset of OAE2 in Core SN°4 (Kuhnt et al., 2017), which is not identified at other localities, was probably associated with local upwelling of cooler deep water masses in the Tarfaya Basin. Cooling during OAE2 occurred in three main steps starting within the onset phase of the positive carbon isotope excursion. The most intense cooling, associated with the Plenus Cold Event, occurred during the peak phase of the excursion (in the trough between the $\delta$13C peaks a and b). The Plenus Cold Event is globally recorded (e.g., Forster et al., 2007; Sinninghe Damsté et al., 2010; Jarvis et al., 2011; Jenkyns et al., 2017) and coincided with invasion of boreal species in the European Chalk Sea (Gale and Christensen, 1996; Voigt et al., 2003), extinction of the planktic foraminifer Rotalipora cushmani (e.g., Kuhnt et al., 2017) and re-oxygenation of bottom water masses (e.g., Eicher and Worstell, 1970; Kuhnt et al., 2005; Friedrich et al., 2006). OAE1a shows a similar response of global temperatures to enhanced organic carbon burial (Kuhnt et al., 2011; Jenkyns, 2018): the main $\delta$18O increase during the latter part of segment C4 in the $\delta$13C curve of Meneghatti et al. (1998) also occurs during the onset phase. A further cooling event within segment C6 follows transient warming during the peak phase. Both cooling events were recognized by Jenkyns (2018) as global events in the northeastern Atlantic Ocean (Naafs and Pancost, 2016), Italy (Bottini et al., 2015), Turkey (Hu et al., 2012) and in the Pacific Ocean (Dumitrescu et al., 2006). Similarities in the $\delta$18O records across both OAEs imply a similar response of the ocean-climate system to lowered atmospheric pCO2-levels due to excess carbon drawdown associated with burial of vast amount of organic material on a global scale.

Line 318: it would be worth looking at the C-segment durations given by Scott , 2016: (Barremian–Aptian–Albian carbon isotope segments as chronostratigraphic signals: numerical age calibration and durations. Stratigraphy, 13, 21–47) to see how they compare with your data.

We have compared our newly reconstructed durations with those from Scott (2016). This comparison is included in Table 2.

Table 2. Durations and sedimentation rates for Aptian C-stages of Menegatti et al. (1998) and comparison with durations of Malinverno et al. (2010) and Scott (2016).

| C-stage | duration (kyr) | durations (kyr) (Malinverno et al., 2010) | durations (kyr) (Scott, 2016) |
|---|---|---|---|
| C8 | 625 | | |
| C7 | 1398 | 1590 | 990 |
| C6 | 315 | 349 | 110 |
| C5 | 281 | 510 | 210 |
| C4 | 388 | 239 | 160 |
| C3 | 434 | 46.7 | 80 |

Line 329: hyphen not necessary in 'orbitally tuned'

Removed

Lines 343–345 and Fig. 2 and Fig. 5: it might be useful to label the features on the OAE 2 carbon-isotope profile (a,b,c,d), as illustrated by Voigt et al., 2017, EPSL. 53, 196–210.

We included the nomenclature of Voigt et al. (2007) in the text and figures to facilitate comparison with global records.

Line 465: 'prevail' - this is present tense and is but one example where past tense should be used for geological narrative. There are many instances of this error in the text. It's also important to maintain clarity when moving from description of an isotope curve to inferences about the environment.

We checked and corrected the manuscript appropriately.

Line 500: compare with the durations given by Scott (see above)

The detailed comparison of the durations of the specific C-stages in section 4.2.1 now includes the durations of Scott (2016). See text below and table 2 above.

Our estimated durations of OAE1a isotope stages agree with those of Malinverno et al. (2010) for the C6 to C8 stages (Table 2), but deviate from the reconstruction of Scott et al. (2016). Minor differences with the estimates of Malinverno et al. (2010) are caused by differing definitions of boundaries between isotope stages in the Cismon core and the LB3/LB1 composite record and by different calculations of orbital periods. The plateau phase (C7) lasted for 1400 kyr in the LB3/LB1 record, in agreement with estimates of 1590 kyr by Malinverno et al. (2010) and of 990 kyr by Scott (2016). The duration of 315 kyr for stage C6 agrees with the 349 kyr estimate by Malinverno et al. (2010), but substantially differ from the 110 kyr proposed by Scott (2016). There are larger deviations from the estimates of Malinverno et al. (2010) and Scott (2016) for stages C5 and C4 in the LB3/LB1 record. Isotope stage C5 has a shorter duration of 280 kyr compared to 510 kyr, estimated by Malinverno et al. (2010), but agrees with the 210 kyr duration of Scott (2016). By contrast, the main increase of the positive carbon isotope excursion, corresponding to C4 (Menegatti et al., 1998), has a duration of 390 kyr in LB3/LB1, which is âĹij60 percent longer than the estimate of 239 kyr by Malinverno et al. (2010) and âĹij140 percent longer than the 160 kyr duration of Scott (2016).

Line 504: change 'In contrast' to 'By contrast'

Changed

Line 552: change 'In contrast' to 'By contrast'

Changed

Lines 579: Mention needs to be made of the key paper by Handoh and Lenton, 2003 (Global biogeochemical Cycles, 17, 1092, who also discuss the cycling of phosphorus to maintain productivity during OAEs. This paper draws on the important papers of Föllmi (Geology, 1995, 23, 503-506; Earth-Science Reviews, 1996, 40, 55–124) that

discuss the long-term stratigraphy of phosphorus in the stratigraphic record.

We added these references in section 4.4. The revised sentence now reads:
Phosphorus is the primary limiting nutrient controlling marine biological productivity on longer (geological) timescales (e.g., Holland, 1978; Broecker and Peng, 1982; Smith, 1984; Codispoti, 1989) with the potential to control the occurrence of high productivity events (e.g., Föllmi 1996; Handoh and Lenton, 2003).

Line 581: say how synthesized from atmospheric nitrogen. This will involve a brief discussion on cyanobacteria and papers by Kuypers et al. (Geology, 2004), and others

A short explanation with appropriate references has been added to section 4.4 as follows:
By contrast to nitrate, which can be synthesized from atmospheric nitrogen primarily by cyanobacterial $N_2$ fixation under anoxic conditions (e.g., Rigby and Batts, 1986; Rau et al., 1987; Kuypers et al., 2004), the phosphorus supply to the ocean is restricted by riverine terrestrial input (Ruttenberg, 2003).

Page 615: is 'largest' the right word? Most significant?

We revised the text as follows:
New high-resolution stable isotope and XRF-scanner data were integrated with published records from Cores LB3 and LB1 in the South Provence Basin (Lorenzen et al., 2013; Moullade et al., 2015) and from Core SN°4 in the Tarfaya Basin (Kuhnt et al., 2017; Beil et al., 2018) to contrast the temporal evolution of two of the most significant Oceanic Anoxic Events: OAE1a and OAE2.

Line 622: given that the durations of the carbon-isotope plateau phases are so different, is their causality different as well? We know that the plateau phase of OAE 2 corresponds with maximum organic-carbon burial, at least in the Tethys–Atlantic region - but there is no such evidence for OAE 1a (except possibly Shatsky Rise). So what is going on?

We addressed this fundamental question by rewriting and expanding sub-section 4.2.1: see above reply to comment 3 concerning the definition of OAE 1a and the abundance of black shales that correspond to the C6 and C7 relatively heavy carbon-isotope segments.

References:

[revised manuscript text omitted]

---

## Author Comment (AC2) · 1 Jan 2020

Reply to Interactive comment on "Cretaceous Oceanic Anoxic Events prolonged by phosphorus cycle feedbacks" by Sebastian Beil et al.

Christian März SC1

c.maerz@leeds.ac.uk

Dear Babette, dear Sebastian and co-authors,

let me first say that I don't tend to write reviews that I haven't been invited to. I do not mean to make the authors' lives harder than they already are. However, the topic your

nice manuscript is about is quite close to my heart, and I have therefore decided to add a few comments that might help to widen the perspective of the manuscript and put into context of a few publications that the authors might have missed. As it happens, some of these publications are (co-) authored by me and my review could be understood as shameless self-promotion. This is not my intention, but the editor might have a different view on this and may therefore decide to ignore my comment.

The manuscript prepared by Beil et al. is an impressive data set on an impressive number samples from two locations that resolve two OAEs (1a and 2) in very high temporal resolution. I have read the comment by Hugh Jenkyns, which focuses on the definition/duration/isotopic expression of the OAEs, and I will not go into any detail on those. Instead, my comment refers to the phosphorus side of the story. I applaude the authors for having generated a very nice P speciation data set, for re- porting the recovery of their extractions relative to total P, and for a very detailed method description in the appendix. In the broadest sense, I also agree with the interpretation of the authors that P recycling from the seafloor during much of OAE2 has potentially led to higher primary productivity, fueling an anoxia-productivity feedback loop that has been previously suggested to extend the "lifetime" of OAEs. My comments, which are all included in the attached PDF as annotations, relate to the (a) a more precise distinction between different redox conditions (namely ferrugi- nous versus euxinic) and (b) the weathering regime. The main reason for raising these issues is that Poulton et al. (2015) conducted a study on the onset of OAE2 from a different Tarfaya core, with a focus on the potential effects of weathering conditions on land on ocean redox, and the related response of the P cycle to these redox changes. Since this manuscript is using very similar methods and proxies on samples from effectively the same location, I think it would be an omission and a missed opportunity to not refer to the published manuscript, and put the new data into context. My comments in the PDF are hopefully self-explanatory, but please feel free to ask for clificaton.

We would first of all like to thank Christian März for helpful, detailed comments that helped us to clarify and improve our manuscript.

We will include a short discussion on the problematic definition of ferruginous in the context of the Cretaceous in section 4.4 (see below). This point was fully discussed by Scholz et al. (2019), who compared iron-speciation proxies in Cretaceous and modern OMZ sediments (Peruvian margin).

An earlier study by Poulton et al. (2015), focusing on Fe speciation proxies during the onset, peak and early plateau phase of OAE2 in nearby drill core S57, found cyclic variations between euxinic and ferruginous conditions. The stratigraphically extended interval (from MCE to early Turonian) investigated at lower resolution by Scholz et al. (2019) is characterized by a high proportion of unpyritized reactive Fe in the total Fe pool. The results of this study are consistent with a proxy signature that is indicative of anoxic and non-sulfidic, so-called ferruginous, water column conditions throughout the studied interval (Poulton and Canfield, 2011). However, by taking into account the low terrigenous sedimentation rates and tropical weathering on the adjacent continent, Scholz et al. (2019) argued that dissolved Fe (and hydrogen sulphide) concentrations in the water column of the Tarfaya system were unlikely higher than those observed in modern upwelling zones (e.g., Peru margin).

Previous studies proposed that phosphorus burial might be enhanced under ferruginous conditions, implying a negative feedback for the oceanic phosphorus pool and primary production (e.g., März et al., 2008). However, Scholz et al. (2019) did not observe a close relationship between Fe and P burial, despite a ferruginous signature in the sediments, which supports the notion that dissolved Fe concentrations and rates of Fe oxide precipitation in the Tarfaya Basin were moderate and overall similar to modern upwelling systems (Wallmann et al., 2019).

I hope the authors will take my relatively minor comments in the good spirit of scientific exchange, and I am looking forward to seeing the final version of the manuscript published in Climate of the Past.

Please also note the supplement to this comment: https://www.clim-pastlines 77-78: Which environments does this refer to? Typically, in sediments underlying most of the oxygenated parts of the world ocean see a "sink switching" not only from organic matter, but also from oxide-bound P to authigenic apatite (see Ruttenberg and Berner, 1993).
As suggested, we revised the text and now include authigenic apatite (Ca-P). We also refer to supplementary material S1 for further information.
Phosphorus remains mainly bound to manganese- and iron-oxides and -hydroxides and occurs as authigenic calcium-bound phosphorus (Supplementary Material S1) in deep-sea sediments underlying well-oxygenated bottom water masses, which typically exhibit C:P below the Redfield ratio.

line 81: I am not sure I would quote this reference as an estimate that is still being used - Ruttenberg's work and especially the discovery of pervasive authigenic apatite formation has superseded this earlier estimate, and I don't think this is being argued with by anyone in the current community.
We acknowledge that the more recent data of Ruttenberg (2003) are now widely accepted. We included the older publication of Broecker and Peng (1982) to underline the point that until recently estimates of the residence time of phosphorus were highly variable. These estimates may still change, as there appear to be imbalances in the phosphorus budget.

lines 93-94: I do not disagree with this statement, but I think the authors should be a bit more cautious regarding the term "anoxic". The increased recycling of P relative to OC from sediments under oxygen-depleted waters is well-documented in many parts of the ocean (nice review paper by Algeo and Ingall, 2001). The formation of phosphorites in the upwelling areas off Peru and Namibia, on the other hand, occurs under quite specific conditions and with the support of specific microbial communities -

and most importantly, under dominantly sulfidic conditions (although the fast changes in bottom water/seafloor redox might also play an important role in enriching P in these shallow environments). In addition, a third line of thought exists regarding the behaviour of P under anoxic, non-sulfidic conditions, which suggests that P can be sequestered into the seafloor under these ferruginous conditions (co-precipitated with Fe minerals or as Fe(II) phosphates). This has been hypothesized for Cretaceous black shales, but also for modern lake sediments, and for subsurface sediments where no sulfide but some dissolved Fe is available. The author won't be surprised that I am raising this point, but I think it is an important one that is well documented in the literature and should be mentioned (even if the authors may come to the conclusion that it is irrelevant in their study).

We agree that complexities of the phosphorus cycle are commonly underestimated. We do not want to discuss in detail the reasons for phosphorus depletion during Cretaceous OAEs, as this would require more extensive data sets. The main aim of this manuscript is to document the availability of the essential nutrient phosphorus and to underline its role in maintaining increased productivity over extended periods of time. A detailed discussion of the mechanisms for increased phosphorus remobilization or non-deposition is beyond the scope of this publication and will be addressed in a future study focused on redox-trends in the Tarfaya Basin (Scholz et al., in prep.).

lines 194-200: Could the authors provide some quality control data for the elements determined (accuracy based on reference materials, precision based on repeat analyses)? I am sure the data are fine, but just to stick to good practice.

The missing values for accuracy and precision based on standards and repeated measurements will be provided in the revised version of the manuscript.

lines 363-366: Are any of these fish remains, nodule, or crusts visible in the core, or do they crop up in the XRF scanning data? If they are, they should be highlighted clearly

as diagenetic features in the data plots - otherwise, it should be mentioned that they were not observed.

We added a sentence clarifying that no fish remains or nodules were visible on the core surfaces nor obvious in the XRF data (see below):

Earlier studies found finely distributed fish debris and fecal pellets in Cenomanian sediments of the Tarfaya Basin (e.g., El Albani et al., 1999) partially reprecipitated as phosphate nodules during early diagenesis (e.g., Leine, 1986; Kuhnt et al., 1997). Phosphatic particles were not observed on the core surfaces and were not apparent in the XRF data of Core SN°4. They were only encountered as minor components in the residues of micropaleontological samples.

lines 370-374: 89 percent is what I would expect as recovery from the chosen extraction technique. But could the authors provide a downcore plot of recovery rates in the Supplement? I am just curious whether this might reveal something about organic P that, even in these old sediments, can still reside in organic matter (after all, the organic matter is still there, in some intervals quite a lot, so it should contain some P as well).

We added a reference to Supplementary Material Figure S16 with the Preact/Ptotal ratio. The overall high ratio of Preact/Ptotal and the increased maturity of sediments from the Tarfaya Basin imply diagenetical sink-switching from organic matter into the more stable phosphorus pools of Ca- and Al/Fe-bound phosphorus.

lines 382-385: Here I would be a little careful regarding anoxic and euxinic conditions. It has been shown by Poulton et al. (2015) that OAE2 at Tarfaya experienced periodic ferruginous conditions; and also Wallmann et al. recently showed independently that ferruginous conditions could be generated in the Cretaeous North Atlantic. In their study, they did not see an increased sequestration of P by Fe-P minerals (different to what Maerz et al., 2008, observed for OAE3 on Demerara Rise). The reasoning

behind this might be quite complex but is related to continental weathering as well as redox conditions and the Fe-C-S cycling on the Tarfaya shelf. I would encourage the author to engage more with that manuscript, especially since it is on material from Tarfaya as well.

See previous reply above concerning the problematic definition of ferruginous in the context of the Cretaceous in section 4.4.

line 425: Shouldn't Corg/Preact be used here?

Corg/Ptotal was intentionally included in Supplementary Figure S16 to show the similar pattern to Corg/Preact, when using total phosphorus concentrations.

lines 471-472: This is at odds with the arrow in Figure 2, which points into the wrong direction for intensified weathering (it's correct in Figure 3).

We corrected the arrow in Figure 2.

lines 475-476: How do you infer that the response to orbital forcing is reduced? There is still a lot of variability in the K/Al record (which is in agreement with the K/Al record in Poulton et al., 2015, who state that orbital pacing is not recorded as clearly in Tarfaya due to the potential for discontinuous sedimentation in shallow waters). It would further be interesting, especially given the very high resolution XRF scanning record, to check if changes in K/Al are correlative with subtle changes in redox conditions, as indicated, for example, by P speciation of the TOC/Rreact ratio.

Figure 2 shows low amplitude variability of the weathering proxy Log(K/Al) during the main phase of OAE2, especially in comparison to the preceding interval, implying low hydrological variability. This dampening suggests a weak response of the hydrological cycle to orbital forcing. Enhanced variability during the plateau phase possibly suggests enhanced response during recovery of the climate-carbon cycle system. We agree that discontinuous sedimentation could erase cyclic pattern in marine

sediments, but no obvious hiatuses are evident in Core SN°4, which would account for the loss of cycles with wavelengths of multiple meters.

lines 587-590: Similar to comment before, this should be visible in the core or other XRF scanning parameters, shouldn't it?
We deleted this sentence, as a discussion on the influence of major sea level variations would be beyond the scope of this manuscript.

lines 607-609: This statement was also made by Poulton et al. (2015), notably during both euxinic and ferruginous intervals that occurred in the early phases of OAE2 at Tarfaya. So apparently no formation of Fe-P minerals that sequestered P during ferruginous intervals on the deeper Demerara Rise.
The high resolution study of Poulton et al. (2015) focused on the onset, peak and early plateau phase of OAE2. By contrast, our lower resolution data set over the mid Cenomanian to early Turonian interval in Core SN°4 allows comparison of background variability with changes occurring during the MCE and OAE2. Our extended data set shows that phosphorus depletion in the Tarfaya Basin exclusively occurred during carbon isotope excursions, which correspond to periods of drastically enhanced organic carbon burial on a global scale. This long-term perspective allows fresh insights into the role of the essential nutrient phosphorus for maintaining increased organic carbon burial over extended periods of time.

Kind regards, Christian Maerz

References:
Broecker, W. S. and Peng, T.-H.: Tracers in the Sea. Eldigio Press, Palisades, New York, USA, 1982.
El Albani, A., Kuhnt, W., Luderer, F., Herbin, J. P., and Caron, M.: Palaeoenvironmental evolution of the Late Cretaceous sequence in the Tarfaya Basin (southwest of Morocco). Geol. Soc. (London) Spec. Publ., 153(1), 223-240, https://doi.org/10.1144/GSL.SP.1999.153.01.14, 1999.

Kuhnt, W., Nederbragt, A., and Leine, L.: Cyclicity of Cenomanian-Turonian organic-carbon-rich sediments in the Tarfaya Atlantic coastal basin (Morocco), Cretaceous Res., 18(4), 587-601, https://doi.org/10.1006/cres.1997.0076, 1997.

Leine, L.: Geology of the Tarfaya oil shale deposit, Morocco, Geol. Mijnbouw, 65, 57-74, 1986.

März, C., Poulton, S. W., Beckmann, B., Küster, K., Wagner, T., and Kasten, S.: Redox sensitivity of P cycling during marine black shale formation: dynamics of sulfidic and anoxic, non-sulfidic bottom waters, Geochim. Cosmochim. Ac., 72(15), 3703-3717, https://doi.org/10.1016/j.gca.2008.04.025, 2008.

Poulton, S. W. and Canfield, D. E.: Ferruginous conditions: a dominant feature of the ocean through Earth's history, Elements, 7(2), 107-112, https://doi.org/10.2113/gselements.7.2.107, 2011.

Poulton, S. W., Henkel, S., März, C., Urquhart, H., Flögel, S., Kasten, S., Siminghe Damste, J. S., and Wagner, T.: A continental-weathering control on orbitally driven redox-nutrient cycling during Cretaceous Oceanic Anoxic Event 2, Geology, 43(11), 963-966, https://doi.org/10.1130/G36837.1, 2015.

Ruttenberg, K. C.: The Global Phosphorus Cycle, in: Treatise on Geochemistry (Vol. 8), edited by Turekian, K. K. and Holland, H. D., Elsevier, 585–643, https://doi.org/10.1016/B0-08-043751-6/08153-6, 2003.

Scholz, F., Beil, S., Flögel, S., Lehmann, M. F., Holbourn, A., Wallmann, K., and Kuhnt, W.: Oxygen minimum zone-type biogeochemical cycling in the Cenomanian-Turonian Proto-North Atlantic across Oceanic Anoxic Event 2. Earth Planet. Sc. Lett., 517, 50-60, https://doi.org/10.1016/j.epsl.2019.04.008, 2019.

Wallmann, K., Flögel, S., Scholz, F., Dale, A. W., Kemena, T. P., Steinig, S., and Kuhnt, W.: Periodic changes in the Cretaceous ocean and climate caused by marine redox

see-saw, Nat. Geosci., 12(6), 456, https://doi.org/10.1038/s41561-019-0359-x, 2019.

---

## Author Comment (AC3) · 1 Jan 2020

Reply to Interactive comment on "Cretaceous Oceanic Anoxic Events prolonged by phosphorus cycle feedbacks" by Sebastian Beil et al.

Matthew Clarkson (Referee) RC2

matthew.clarkson@erdw.ethz.ch

The manuscript presents an impressive dataset of P-speciation data and high resolution XRF core scans, which help build on earlier works regarding i) the duration of OAEs, and ii) the hypothesis for P-cycling as an important feedback mechanism for

OAE development. The work involved in this manuscript could feasibly represent two papers, if the authors saw fit, as the cyclo-stratigraphy aspect over-shadows the P spe- ciation and the data presentation become very lengthy. I have read through the detailed comments from Hugh Jenkyns and Cristian Maerz and agree with their inputs. I will try to give additional contributions, rather than repeating their observations. Generally, this is an impressive dataset and it shouldn't take much work to address these comments.

We would like to thank Matthew Clarkson for his helpful, constructive comments and suggestions, which we have followed as much as possible.

General Comments:

I think there is a missed opportunity here in that one of these cores has been exten- sively studied previously by the authors (Scholz et al., 2019), with Fe-speciation, redox sensitive metals (Mo, V) and N isotopes. The new P-speciation data would comple- ment this previous study very nicely and more could be made of integrating the two datasets. This could be valuable for the discussion of redox and P cycling through OAE2 and would help give more contextual information, particularly with reference to the evolving nature of redox conditions through the core. I think it would be very useful to the community to examine P-speciation results within the context of the established redox framework that varies locally from nitrogenous to euxinic, and compare this to intervals of ferruginous and euxinic deposition elsewhere (Poulton et al., 2015), and so I am somewhat mirroring a comment made by Dr. Maerz.

A preliminary discussion of the long-term redox change in the Tarfaya Basin was in- cluded in Beil et al. (2018). We feel that a more detailed discussion of redox-conditions is beyond the scope of the present paper, which focuses on a synthetic comparison of OAE1a and OAE2. However, a detailed discussion of redox-changes is in preparation (Scholz et al., in prep.) with principal aim to reconstruct redox-variability at high

resolution in the Tarfaya Basin.

Minor comments:

Line 93: 'oceanic anoxia'
Changed

Line 114: The MCE is referred to frequently as it appears in the records, however not much background is given on the significance of this event. Please detail if this is a local feature or a global event comparable to the other OAEs studied.
We propose to add the following text to the first paragraph of the introduction:
The mid-Cenomanian event (MCE; Coccioni and Galeotti, 2003) appears to represent a less intense precursor event of OAE2. However, detailed records of the MCE are still sparse with most studies focusing on the higher amplitude events (OAE1a, b, c, d and OAE2). Records until now are predominantly from the North Atlantic and Tethys region (e.g., Umbria Marche Basin (Coccioni and Galeotti, 2003); English Chalk (Gale, 1989; Jenkyns et al., 1994); Western Interior Seaway (Keller et al., 2004); Blake Nose (Ando et al., 2009))., which display a positive isotope excursion during the Thalmanninella reicheli foraminiferal zone. In shelf areas of the global ocean, major sea level changes associated with the cycles Ce2.1 and Ce3 of Gale et al. (2002) may have caused long lasting hiatuses obliterating evidence of the MCE.

Line 160: As a disclaimer, I am not so familiar with XRF core scanning techniques, but I would be suspicious of using Fe as a terrestrial element, included in the logTerr/Ca proxy, as there is likely redox-dependent behaviour in these settings that would obscure or bias trends in terrestrial elements if Fe is included. It might be that Fe is lost from the sediment due to reduction in the pore-waters (thereby removing any Fe

cycles), or that Fe has been enriched through Fe-shuttling across the basin. It would be possible that the stepped increase in Terr/Ca could be caused by an increase in Fe, due to enrichment of highly reactive Fe-phases (e.g. at the onset of OAE1a). It is also possible that this could create apparent cyclicity, analogous to the cyclicity in FeHR/FeT in other Tarfaya data (Poutlon et al, 2015). If Fe is plotted separately or removed from this measure, do you see any behaviour that might be indicative of local redox changes dominating the record?

This could be an opportunity to add additional information on redox systematics. Can you pull out Fe/Al from the XRF data to aid comparison to the Fe-speciation cyclicity ob- served by Poulton et al., 2015 in the other Tarfaya core and the previous Fe-speciation data of Scholz et al., 2019?

We added the following figure to the supplementary material showing Log(Terr/Ca) calculated with (red) and without (black) iron for OAE1a and OAE2. This figure reveals no major deviations between datasets.

No significant influence of redox-variability implies a predominantly detritic reservoir for iron. Fe/Al (not shown) is therefore controlled by the composition of deposited terrigenous material and cannot be used as a proxy for redox changes in both basins. A detailed discussion of redox-conditions is beyond the scope of the present paper, but a detailed discussion of redox-changes is in preparation with principal aim to reconstruct redox-variability at high resolution in the Tarfaya Basin.

Also, what about the dilution effect of Ca from high organic carbon production, would this potentially create cycles or stepped changes through the OAEs. There seems to cycles in TOC from just looking at the linescan photograph, so how much of the cyclicity in logTerr/Ca can be explained by simply changing CaCO3 concentration?

Yes, Log(Terr/Ca) is a proxy for carbonate content and shows a good correlation with carbonate content as shown in Beil et al. (2018). We assume that most of

this variability is rooted in the fluctuating terrigenous input associated with changing conditions on land.

Could you also please clarify what NGR represents in terms of sedimentary components that drives the cyclicity, and how this links to the orbital pacing mechanisms.
We expanded section 2.2 as follows:
The intensity of natural gamma radiation is predominantly influenced by the concentration of three different elements: Potassium (K), Uranium (U) and Thorium (Th). All three elements are bound to clay minerals with uranium also adhesively enriched in organic matter. In environments with low terrigenous input and high organic matter deposition as at the palaeo-position of Core SN°4, NGR is predominantly controlled by the concentration of organic matter. Predominant sedimentary control can be assumed for Cores LB3/LB1 characterized by low organic matter content.

Line 196: is smoked the correct term for this? ashed?
Smoked is the correct terminology: the liquid was minimized by evaporation.

Line 280: the PCE is often associated with faunal changes that represent different water mass movements or local re-oxygenation. I think it is a bit misleading to focus on the extinction aspect. More could be done to reference other studies here.
We expanded chapter 3.2 to include the most important of the environmental changes registered worldwide as follows:
The Plenus Cold Event is globally recorded (Forster et al., 2007; Sinninghe Damsté et al., 2010; Jarvis et al., 2011; Jenkyns et al., 2017) and coincided with invasion of boreal species in the European Chalk Sea (Gale and Christensen, 1996; Voigt et al., 2003), extinction of the planktic foraminifer Rotalipora cushmani (e.g., Kuhnt et al., 2017) and re-oxygenation of bottom water masses (e.g., Eicher and Worstell, 1970; Kuhnt et al., 2005; Friedrich et al., 2006).

References:

[revised manuscript text omitted]

---

## Author Response (AR1)

This paper aims to illustrate the potential role of P-cycle feedbacks in prolonging OAE 1a and OAE 2. The paper contains useful data and some excellent diagrams but is rather densely written, skips over some important problems, switches tenses a lot when describing geological phenomena, and ignores some relevant literature. The fundamental point that the low P/TOC ratios in the OAE sediments points definitively to phosphorus recycling (and nutrient re-supply to planktonic biota) during these events tends is easily lost. The issue of P recycling during OAEs has, of course, been made previously, including from a modelling perspective (e.g. Mort papers; Nederbragt et al). The value of the account lies in the fact that the sections described are stratigraphically very expanded and give superb detail as to changes in the carbon cycle before, during and after OAEs.

First of all, we would like to sincerely thank Hugh Jenkyns for his insightful and constructive feedback. Following his comments, we revised and streamlined the manuscript to improve readability and highlight the main findings of our study. We also addressed important questions, which were not previously touched upon, and we included missing essential references.

Abstract and beyond: the statement that the evolution of the carbon-isotope curve of the two OAEs, as classically defined, shows remarkable similarities needs to be qualified.
We discuss in more detail the similarities and dissimilarities in the evolution of the carbon-isotope curves of the two OAEs in the revised manuscript. We modified the abstract and expanded section 3.1 (lines 277-298).

The defining characteristic of OAE 2 is the overarching positive excursion; for OAE 1a it's the negative excursion. Many OAE 2 sequences (e.g. Eastbourne, UK) show no negative excursion, although its absence is probably due to the presence of the sub-plenus erosion surface in the case of the English section. More needs to be made of all this because the apparently more stratigraphically complete Tarfaya record of OAE 2 clearly offers a unique perspective. The New Zealand record of Gangl et al. (EPSL, 518, 172–182) and Japanese record of Nemoto and Hasegawa (Palaeo-cubed, 309, 271–280) may also show this negative excursion but it is certainly not everywhere apparent.
To address this comment, we inserted a new paragraph in section 4.2.2 of the discussion (lines 532-544).

As regards OAE 1a, as illustrated in Fig. 4, the main positive excursion extends higher than the C6 segment (1.e. post OAE 1a - unless C6 is extended higher in the section). Do we need a total redefinition of OAE 1a, as implied here? If so, all of this needs to be made clear as perhaps we have been biased by the records of the Cismon and Piobbico cores. But there is a problem: where are the abundant black

shales that correspond to the C6 and C7 relatively heavy carbon-isotope segments, given that the original OAE definition is rooted in the quasi-coeval organic-rich record on a global basis?

We agree that this issue needs to be further discussed. We expanded the discussion on the discrepancy between the stratigraphic extension of OAE1a black shale occurrences and the positive carbon isotope excursion (section 4.2.1; lines 579-511).

Line 21: not clear which events are being referred to with 'respectively'
We deleted this word and revised the sentence (lines 17-20).

Lines 23–25: nutrients may have been supplied by basalt–seawater interaction, probably involving LIPs. (Mentioned later in the text but not here)
We now mention this possible nutrient source in lines 21-24.

Line 55: cite original paper by Scholle and Arthur (1980)
We added this reference in line 61.

Line 66: Are these Mort papers the appropriate references for discussion of transgression? See Jenkyns (1980)
We included the primary citation of Jenkyns (1980) and we extended the sentence to include the additional source of increased terrestrial weathering (lines 68-73).

Line 92: rewrite as: 'A variety of phosphorus species are discriminated against in these sediments.
We revised the sentence as suggested.

Line 98: change 'In contrast' to 'By contrast'
We revised the sentence as suggested.

Line 131: hyphenate 'intermediate-resolution' to read as written here
Changed

Line 164: do you mean nannofossils and planktonic foraminifera? 'Shells' rather implies macrofossils.
We revised the sentence in lines 172-174.

Line 189: hyphenate 'metal-free' to read as written here

Changed

Line 271: change 'In contrast' to 'By contrast'

Changed

Line 280: state in which segments of the OAE 1a record the cooling events have been identified. Do they conform to those illustrated in Jenkyns, 2018 (Phil .Trans

Roy. Soc.) from multiple localities, namely: C3, C4 and C6? Which cooling events in the OAE 2 record correspond with the Plenus Cold Event? Are these multiple events registered anywhere else? Do they relate to the fact that Tarfaya was a palaeo-upwelling site with upward movements of cooler water or are they global? The largest positive oxygen- isotope shift (Fig. 2) seems to predate the rise in carbon isotopes: i.e. before major global carbon burial was registered, which is not as stated in the text (line 284).

The major cooling events that occurred during C4 and C6 correspond to the global events illustrated by Jenkyns (2018). A minor cooling of probable regional character (Jenkyns, 2018) is also evident during stage C3. We expanded section 3.2 and added relevant references (lines 300-317).

Line 318: it would be worth looking at the C-segment durations given by Scott , 2016: (Barremian–Aptian–Albian carbon isotope segments as chronostratigraphic signals: numerical age calibration and durations. Stratigraphy, 13, 21–47) to see how they compare with your data.

We have compared our newly reconstructed durations with those from Scott (2016). This comparison is included in Table 2.

Line 329: hyphen not necessary in 'orbitally tuned'

Removed

Lines 343–345 and Fig. 2 and Fig. 5: it might be useful to label the features on the OAE 2 carbon-isotope profile (a,b,c,d), as illustrated by Voigt et al., 2017, EPSL. 53, 196–210.

We included the nomenclature of Voigt et al. (2007) in the text and figures to facilitate comparison with global records.

Line 465: 'prevail' - this is present tense and is but one example where past tense should be used for geological narrative. There are many instances of this error in the text. It's also important to maintain clarity when moving from description of an isotope curve to inferences about the environment.

We checked and corrected the manuscript appropriately.

Line 500: compare with the durations given by Scott (see above)

The detailed comparison of the durations of the specific C-stages in section 4.2.1 now includes the durations of Scott (2016) in table 2 and lines 512-522.

Line 504: change 'In contrast' to 'By contrast'

Changed

Line 552: change 'In contrast' to 'By contrast'

Changed

Lines 579: Mention needs to be made of the key paper by Handoh and Lenton, 2003 (Global biogeochemical Cycles, 17, 1092, who also discuss the cycling of phosphorus to maintain productivity during OAEs. This paper draws on the important papers of Föllmi (Geology, 1995, 23, 503-506; Earth-Science Reviews,

1996, 40, 55–124) that discuss the long-term stratigraphy of phosphorus in the stratigraphic record.
We added these references in section 4.4 in line 622.

Line 581: say how synthesized from atmospheric nitrogen. This will involve a brief discussion on cyanobacteria and papers by Kuypers et al. (Geology, 2004), and others
A short explanation with appropriate references has been added to section 4.4 in lines 622-624.

Page 615: is 'largest' the right word? Most significant?
We revised the text in lines 668-671.

Line 622: given that the durations of the carbon-isotope plateau phases are so different, is their causality different as well? We know that the plateau phase of OAE 2 corresponds with maximum organic-carbon burial, at least in the Tethys–Atlantic region - but there is no such evidence for OAE 1a (except possibly Shatsky Rise). So what is going on?
We addressed this fundamental question by rewriting and expanding sub-section 4.2.1 (lines 479-511).

**Reply to *Interactive comment on* "Cretaceous Oceanic Anoxic Events prolonged by phosphorus cycle feedbacks" *by* Sebastian Beil et al.**

**Christian März SC1**

c.maerz@leeds.ac.uk

Dear Babette, dear Sebastian and co-authors,

let me first say that I don't tend to write reviews that I haven't been invited to. I do not mean to make the authors' lives harder than they already are. However, the topic your nice manuscript is about is quite close to my heart, and I have therefore decided to add a few comments that might help to widen the perspective of the manuscript and put into context of a few publications that the authors might have missed. As it happens, some of these publications are (co-) authored by me and my review could be understood as shameless self-promotion. This is not my intention, but the editor might have a different view on this and may therefore decide to ignore my comment.

The manuscript prepared by Beil et al. is an impressive data set on an impressive number samples from two locations that resolve two OAEs (1a and 2) in very high temporal resolution. I have read the comment by Hugh Jenkyns, which focuses on the definition/duration/isotopic expression of the OAEs, and I will not go into any detail on those. Instead, my comment refers to the phosphorus side of the story.

I applaude the authors for having generated a very nice P speciation data set, for re-porting the recovery of their extractions relative to total P, and for a very detailed method description in the appendix. In the broadest sense, I also agree with the interpretation of the authors that P recycling from the seafloor during much of OAE2 has potentially led to higher primary productivity, fueling an anoxia-productivity feedback loop that has been previously suggested to extend the "lifetime" of OAEs.

My comments, which are all included in the attached PDF as annotations, relate to the (a) a more precise distinction between different redox conditions (namely ferrugi- nous versus euxinic) and (b) the weathering regime. The main reason for raising these issues is that Poulton et al. (2015) conducted a study on the onset of OAE2 from a different Tarfaya core, with a focus on the potential effects of weathering conditions on land on ocean redox, and the related response of the P cycle to these redox changes. Since this manuscript is using very similar methods and proxies on samples from effectively the same location, I think it would be an omission and a missed opportunity to not refer to the published manuscript, and put the new data into context. My comments in the PDF are hopefully self-explanatory, but please feel free to ask for clarificaton.

We would first of all like to thank Christian März for helpful, detailed comments that helped us to clarify and improve our manuscript.

We included a short discussion on the problematic definition of ferruginous in the context of the Cretaceous in section 4.4 (lines 646-658). This point was fully discussed by Scholz et al. (2019), who compared iron-speciation proxies in Cretaceous and modern OMZ sediments (Peruvian margin).

I hope the authors will take my relatively minor comments in the good spirit of scientific exchange, and I am looking forward to seeing the final version of the manuscript published in Climate of the Past.

Please also note the supplement to this comment:

https://www.clim-past-discuss.net/cp-2019-118/cp-2019-118-SC1-supplement.pdf

lines 77-78: Which environments does this refer to? Typically, in sediments underlying most of the oxygenated parts of the world ocean see a "sink switching" not only from organic matter, but also from oxide-bound P to authigenic apatite (see Ruttenberg and Berner, 1993).

As suggested, we revised the text and now include authigenic apatite (Ca-P) (lines 81-91). We also refer to supplementary material S1 for further information.

line 81: I am not sure I would quote this reference as an estimate that is still being used - Ruttenberg's work and especially the discovery of pervasive authigenic apatite formation has superseded this earlier estimate, and I don't think this is being argued with by anyone in the current community.

We acknowledge that the more recent data of Ruttenberg (2003) are now widely accepted. We included the older publication of Broecker and Peng (1982) to underline the point that until recently estimates of the residence time of phosphorus were highly variable. These estimates may still change, as there appear to be imbalances in the phosphorus budget.

lines 93-94: I do not disagree with this statement, but I think the authors should be a bit more cautious regarding the term "anoxic". The increased recycling of P relative to OC from sediments under oxygen-depleted waters is well-documented in many parts of the ocean (nice review paper by Algeo and Ingall, 2001). The formation of phosphorites in the upwelling areas off Peru and Namibia, on the other hand, occurs under quite specific conditions and with the support of specific microbial communities - and most importantly, under dominantly sulfidic conditions (although the fast changes in bottom water/seafloor redox might also play an important role in enriching P in these shallow environments).

In addition, a third line of thought exists regarding the behaviour of P under anoxic, non-sulfidic conditions, which suggests that P can be sequestered into the seafloor under these ferruginous conditions (co-precipitated with Fe minerals or as Fe(II) phosphates). This has been hypothesized for Cretaceous black shales, but also for modern lake sediments, and for subsurface sediments where no sulfide but some dissolved Fe is available. The author won't be surprised that I am raising this point, but I think it is an important one that is well documented in the literature and should be mentioned (even if the authors may come to the conclusion that it is irrelevant in their study).

We agree that complexities of the phosphorus cycle are commonly underestimated. We do not want to discuss in detail the reasons for phosphorus depletion during Cretaceous OAEs, as this would require more extensive data sets. The main aim of this manuscript is to document the availability of the essential nutrient phosphorus and to underline its role in maintaining increased productivity over extended periods of time. A detailed discussion of the mechanisms for increased phosphorus remobilization or non-deposition is beyond the scope of this publication and will be addressed in a future study focused on redox-trends in the Tarfaya Basin (Scholz et al., in prep.).

lines 194-200: Could the authors provide some quality control data for the elements determined (accuracy based on reference materials, precision based on repeat analyses)? I am sure the data are fine, but just to stick to good practice.

The missing values for accuracy and precision based on standards and repeated

measurements are now provided in lines 231-232.

lines 363-366: Are any of these fish remains, nodule, or crusts visible in the core, or do they crop up in the XRF scanning data? If they are, they should be highlighted clearly as diagenetic features in the data plots - otherwise, it should be mentioned that they were not observed.

We added a sentence clarifying that no fish remains or nodules were visible on the core surfaces nor obvious in the XRF data in lines 375-377.

lines 370-374: 89 percent is what I would expect as recovery from the chosen extraction technique. But could the authors provide a downcore plot of recovery rates in the Supplement? I am just curious whether this might reveal something about organic P that, even in these old sediments, can still reside in organic matter (after all, the organic matter is still there, in some intervals quite a lot, so it should contain some P as well).

We added a reference to Supplementary Material Figure S7.1 in line 382 with the $P_{react}/P_{total}$ ratio. The overall high ratio of $P_{react}/P_{total}$ and the increased maturity of sediments from the Tarfaya Basin imply diagenetical sink-switching from organic matter into the more stable phosphorus pools of Ca- and Al/Fe-bound phosphorus.

lines 382-385: Here I would be a little careful regarding anoxic and euxinic conditions. It has been shown by Poulton et al. (2015) that OAE2 at Tarfaya experienced periodic ferruginous conditions; and also Wallmann et al. recently showed independently that ferruginous conditions could be generated in the Cretaeous North Atlantic. In their study, they did not see an increased sequestration of P by Fe-P minerals (different to what Maerz et al., 2008, observed for OAE3 on Demerara Rise). The reasoning behind this might be quite complex but is related to continental weathering as well as redox conditions and the Fe-C-S cycling on the Tarfaya shelf. I would encourage the author to engage more with that manuscript, especially since it is on material from Tarfaya as well.

See previous reply above concerning the problematic definition of ferruginous in the context of the Cretaceous in section 4.4.

line 425: Shouldn't Corg/Preact be used here?

$C_{org}/P_{total}$ was intentionally included in Supplementary Figure S7.1 to show the similar pattern to $C_{org}/P_{react}$, when using total phosphorus concentrations.

lines 471-472: This is at odds with the arrow in Figure 2, which points into the wrong direction for intensified weathering (it's correct in Figure 3).

We corrected the arrow in Figure 2.

lines 475-476: How do you infer that the response to orbital forcing is reduced? There is still a lot of variability in the K/Al record (which is in agreement with the K/Al record in Poulton et al., 2015, who state that orbital pacing is not recorded as clearly in Tarfaya due to the potential for discontinuous sedimentation in shallow waters). It would further be interesting, especially given the very high resolution XRF scanning record, to check if changes in K/Al are correlative with subtle changes in redox conditions, as indicated, for example, by P speciation of the TOC/Rreact ratio.

Figure 2 shows low amplitude variability of the weathering proxy Log(K/Al) during the main phase of OAE2, especially in comparison to the preceding interval, implying low hydrological variability. This dampening suggests a weak response of the hydrological cycle to orbital forcing. Enhanced variability during the plateau phase possibly suggests enhanced response during recovery of the climate-carbon cycle system. We agree that discontinuous sedimentation could erase cyclic pattern in marine sediments, but no obvious hiatuses are evident in Core SN°4, which would account for the loss of cycles with wavelengths of multiple meters.

lines 587-590: Similar to comment before, this should be visible in the core or other XRF scanning parameters, shouldn't it?

We deleted this sentence, as a discussion on the influence of major sea level variations would be beyond the scope of this manuscript.

lines 607-609: This statement was also made by Poulton et al. (2015), notably during both euxinic and ferruginous intervals that occurred in the early phases of OAE2 at Tarfaya. So apparently no formation of Fe-P minerals that sequestered P during ferruginous intervals on the deeper Demerara Rise.

The high resolution study of Poulton et al. (2015) focused on the onset, peak and early plateau phase of OAE2. By contrast, our lower resolution data set over the mid Cenomanian to early Turonian interval in Core SN°4 allows comparison of background variability with changes occurring during the MCE and OAE2. Our extended data set shows that phosphorus depletion in the Tarfaya Basin exclusively occurred during carbon isotope excursions, which correspond to periods of drastically enhanced organic carbon burial on a global scale. This long-term perspective allows fresh insights into the role of the essential nutrient phosphorus for maintaining increased organic carbon burial over extended periods of time.

Kind regards, Christian Maerz

**Reply to *Interactive comment on* "Cretaceous Oceanic Anoxic Events prolonged by phosphorus cycle feedbacks" *by* Sebastian Beil et al.**

**Matthew Clarkson (Referee) RC2**

matthew.clarkson@erdw.ethz.ch

The manuscript presents an impressive dataset of P-speciation data and high reso- lution XRF core scans, which help build on earlier works regarding i) the duration of OAEs, and ii) the hypothesis for P-cycling as an important feedback mechanism for OAE development. The work involved in this manuscript could feasibly represent two papers, if the authors saw fit, as the cyclo-stratigraphy aspect over-shadows the P spe- ciation and the data presentation become very lengthy. I have read through the detailed comments from Hugh Jenkyns and Cristian Maerz and agree with their inputs. I will try to give additional contributions, rather than repeating their observations. Generally, this is an impressive dataset and it shouldn't take much work to address these comments.

We would like to thank Matthew Clarkson for his helpful, constructive comments and suggestions, which we have followed as much as possible.

General Comments:

I think there is a missed opportunity here in that one of these cores has been exten- sively studied previously by the authors (Scholz et al., 2019), with Fe-speciation, redox sensitive metals (Mo, V) and N isotopes. The new P-speciation data would comple- ment this previous study very nicely and more could be made of integrating the two datasets. This could be valuable for the discussion of redox and P cycling through OAE2 and would help give more contextual information, particularly with reference to the evolving nature of redox conditions through the core. I think it would be very useful to the community to examine P-speciation results within the context of the established redox framework that varies locally from nitrogenous to euxinic, and compare this to intervals of ferruginous and euxinic deposition elsewhere (Poulton et al., 2015), and so I am somewhat mirroring a comment made by Dr. Maerz.

A preliminary discussion of the long-term redox change in the Tarfaya Basin was included in Beil et al. (2018). We feel that a more detailed discussion of redox-conditions is beyond the scope of the present paper, which focuses on a synthetic comparison of OAE1a and OAE2. However, a detailed discussion of redox-changes is in preparation (Scholz et al., in prep.) with principal aim to reconstruct redox-variability at high resolution in the Tarfaya Basin.

Minor comments:

Line 93: 'oceanic anoxia'

Changed

Line 114: The MCE is referred to frequently as it appears in the records, however not much background is given on the significance of this event. Please detail if this is a local feature or a global event comparable to the other OAEs studied.

We added a short paragraph about the MCE to the introduction (lines 40-47).

Line 160: As a disclaimer, I am not so familiar with XRF core scanning techniques, but I would be suspicious of using Fe as a terrestrial element, included in the logTerr/Ca proxy, as there is likely redox-dependent behaviour in these settings that would obscure or bias trends in terrestrial elements if Fe is included. It might be that Fe is lost from the sediment due to reduction in the pore-waters (thereby removing any Fe cycles), or that Fe has been enriched through Fe-shuttling across the basin. It would be possible that the stepped increase in Terr/Ca could be caused by an increase in Fe, due to enrichment of highly reactive Fe-phases (e.g. at the onset of OAE1a). It is also possible that this could create apparent cyclicity, analogous to the cyclicity in FeHR/FeT in other Tarfaya data (Poutlon et al, 2015). If Fe is plotted separately or removed from this measure, do you see any behaviour that might be indicative of local redox changes dominating the record?

This could be an opportunity to add additional information on redox systematics. Can you pull out Fe/Al from the XRF data to aid comparison to the Fe-speciation cyclicity ob- served by Poulton et al., 2015 in the other Tarfaya core and the previous Fe-speciation data of Scholz et al., 2019?

We added a short sentence to the introduction (lines 177-179) and figure S2.1 to the supplementary material showing Log(Terr/Ca) calculated with (red) and without (black) iron for OAE1a and OAE2. This figure reveals no major deviations between datasets.

No significant influence of redox-variability implies a predominantly detritic reservoir for iron. Fe/Al (not shown) is therefore controlled by the composition of deposited terrigenous material and cannot be used as a proxy for redox changes in both basins. A detailed discussion of redox-conditions is beyond the scope of the present paper, but a detailed discussion of redox-changes is in preparation with principal aim to reconstruct redox-variability at high resolution in the Tarfaya Basin.

Also, what about the dilution effect of Ca from high organic carbon production, would this potentially create cycles or stepped changes through the OAEs. There seems to cycles in TOC from just looking at the linescan photograph, so how much of the cyclicity in logTerr/Ca can be explained by simply changing CaCO3 concentration?

Yes, Log(Terr/Ca) is a proxy for carbonate content and shows a good correlation with carbonate content as shown in Beil et al. (2018). We assume that most of this variability is rooted in the fluctuating terrigenous input associated with changing conditions on land.

Could you also please clarify what NGR represents in terms of sedimentary compo- nents that drives the cyclicity, and how this links to the orbital pacing mechanisms.

We expanded section 2.2 (lines 148-154) for a brief discussion of the influence of the different sedimentary components on the NGR records in the South Provence and Tarfaya Basins.

Line 196: is smoked the correct term for this? ashed?

Smoked is the correct terminology: the liquid was minimized by evaporation.

Line 280: the PCE is often associated with faunal changes that represent different water mass movements or local re-oxygenation. I think it is a bit

misleading to focus on the extinction aspect. More could be done to reference other studies here.

We expanded chapter 3.2 (lines 304-310) to include the most important of the environmental changes registered worldwide.

[revised manuscript text omitted]

hat gelöscht: in the increase

hat gelöscht: 11

hat gelöscht: wo of these peaks …e maxima at 220.19 and 104.40 m are …lso characterized by…xhibit high $P_{total}$ concentrations and maxima in the …total/Al maximaratio… The first peak at 220.19 m precedes the first $\delta^{13}C_{org}$ increase in $\delta^{13}C_{org}$ …

hat gelöscht: Fig. S16

hat gelöscht: of …or the deposited

hat gelöscht: Fig. S17…upplementary Material S7.1) during OAE2 postdates the increase in the atomic $C_{org}/N_{total}$-ratio, interpreted by Beil et al. (2018) as enhanced cycling of nitrogen rich organic matter within a dysoxic or anoxic water column. The increase in $C_{org}/P_{total}$ starts immediately above the prominent peak in Ca-/AlFe-bound and total phosphorus and coincides with the first $\delta^{13}C_{org}$ maximum of OAE2 (Figs. 6 and 7). The highest $C_{org}/P_{total}$ ratio is determined at 90.78 m within the plateau phase of OAE2. This peak coincides with minima in $P_{total}$ and $P_{total}$/Al, implying remobilization either synsedimentary due to preferential recycling of phosphorus in the water column and/or at

hat gelöscht: are shown in… …ig. 7) and compared to the respective Redfield ratio (Redfield, 1958; Redfield et al., 1963) representative of the mean composition of marine phytoplankton assumed to represent the main source of organic matter and sedimentary phosphorus to the sediments. Both ratios …re always lower or close to the Redfield ratio (C:N:P =of

hat gelöscht: ; Redfield et al.

hat gelöscht: for two intervals. The first falls within…uring the MCE and OAE2, when $C_{org}/P_{react}$ is equal to or higher than 106:1 and $N_{total}/P_{react}$ is close to the predicted ratio of 16:1. The second interval starts during OAE2… above the prominent peak in $P_{total}$, synchronous with the Transgressive Surface (TS) of Ce5. …oth ratios surpass predicted values at the first maximum in $\delta^{13}C_{org}$ peak of OAE2, decrease slightly during the…he Plenus Cold Event and show a large increase during the Plateau plateau Phase…hase. The remaining part of the sampled …nterval is characterized by increased values above the…background level that is characteristic for…s in the lower part of Core SN°4 until

hat formatiert: Nicht Hervorheben

hat gelöscht: Three different intervals across OAE2 are characterized by different phosphorus accumulation

hat gelöscht: Across OAE2, a…ccumulation rates (AR) of specifically $P_{react}$, but also $P_{total}$, Al/Fe-bound P and Ca-bound P decline during the onset, peak and plateau phase of OAE2 range between …

hat formatiert:

hat gelöscht: .

hat formatiert:

hat formatiert: Nicht Hervorheben

[revised manuscript text omitted]

**hat gelöscht:** The similarities in the general shape of the $\delta^{13}C$ excursion of OAE1a and OAE2 (precursor, onset, peak and plateau phase) suggest similar forcing and response mechanisms. However, there are also remarkable differences in the amplitude and duration of individual phases: in particular, the higher amplitude and extended duration of the precursor phase (negative $\delta^{13}C$ excursion preceding the onset of the positive $\delta^{13}C$ excursion) and the exceptionally long duration of the plateau phase of OAE1a, which in most classic localities is not associated with the deposition of organic carbon rich black shales ("Selli level"). The different durations of the precursor and plateau phases of OAE1a and OAE2 may have been linked to the magnitude and duration of the triggering volcanic exhalations. In addition, different orbital configurations may have influenced long-term marine organic carbon burial on a global scale. Furthermore, obliquity-forced intensification of monsoonal systems may have resulted in periods of enhanced tropical weathering associated with nutrient supply to the ocean and wind driven equatorial upwelling, which promoted carbon sequestration during the plateau and recovery phases. Periods of low 41 kyr variability in orbital obliquity (obliquity nodes), which occur every 1.2 Myr and are commonly associated with global cooling episodes in Cenozoic warm climate records (e.g., Pälike et al., 2006), may have triggered interruptions or termination of globally enhanced carbon burial.¶

**hat gelöscht:** 4

**hat formatiert:** Nicht Hervorheben

**hat formatiert:** Nicht Hervorheben

**hat formatiert:** Nicht Hervorheben

**hat gelöscht:** 3

**hat gelöscht:** shallow

**hat gelöscht:** In

**hat gelöscht:** y

**hat formatiert:** Nicht Hervorheben

[revised manuscript text omitted]